# Symmetries, Conservation Laws and Entanglement in Non-Hermitian Fermionic Lattices

R. D. Soares[1,2], Y. Le Gal[1], C. Y. Leung[3], D. Meidan[4,5], A. Romito[3], M. Schirò[1]

**1** JEIP, UAR 3573 CNRS, Collège de France, PSL Research University, 11 Place Marcelin Berthelot, 75321 Paris Cedex 05, France
**2** Max Planck Institute for the Physics of Complex Systems, Nöthnitzer Str. 38, 01187 Dresden, Germany
**3** Department of Physics, Lancaster University, United Kingdom
**4** Department of Physics, Ben-Gurion University of the Negev, Beer-Sheva 84105, Israel
**5** SPEC, CEA, CNRS, Université Paris-Saclay, 91191 Gif-sur-Yvette, France

April 14, 2025

## Abstract

Non-Hermitian quantum many-body systems feature steady-state entanglement transitions driven by the competition between unitary dynamics and dissipation. In this work, we reveal the fundamental role of conservation laws in shaping this competition. Focusing on translation-invariant non-interacting fermionic models with U(1) symmetry, we present a theoretical framework to understand the structure of the steady-state of these models and their entanglement content based on two ingredients: the nature of the spectrum of the non-Hermitian Hamiltonian and the constraints imposed on the steady-state single-particle occupation by the conserved quantities. These emerge from an interplay between Hamiltonian symmetries and initial state, due to the non-linearity of measurement back-action. For models with complex energy spectrum, we show that the steady state is obtained by filling single-particle right eigenstates with the largest imaginary part of the eigenvalue. As a result, one can have partially filled or fully filled bands in the steady-state, leading to an entanglement entropy undergoing a filling-driven transition between critical sub-volume scaling and area-law, similar to ground-state problems. Conversely, when the spectrum is fully real, we provide evidence that local observables can be captured using a diagonal ensemble, and the entanglement entropy exhibits a volume-law scaling independently on the initial state, akin to unitary dynamics. We illustrate these principles in the Hatano-Nelson model with periodic boundary conditions and the non-Hermitian Su-Schrieffer-Heeger model, uncovering a rich interplay between the single-particle spectrum and conservation laws in determining the steady-state structure and the entanglement transitions. These conclusions are supported by exact analytical calculations and numerical calculations relying on the Faber polynomial method.

# 1 Introduction

Recent years have seen major progresses in the understanding of quantum dynamics in closed many-body systems [1, 2]. Here, the unitarity of time evolution puts strong constraints on the way correlations and entanglement spread through the system [3, 4], or local observables reach a stationary state and possibly approach thermal equilibrium at long-times [5].

A novel frontier of many-body quantum dynamics concerns the role of dissipation due to coupling to external environments, leading to a non-unitary time evolution [6]. An example of non-unitary dynamics which has attracted widespread interest is provided by non-Hermitian Hamiltonians [7], which arise naturally in the description of open quantum systems as no-click limit of a monitored evolution [8–10] and in the context of dynamics of systems with loss and gain [11–13].

Non-Hermitian quantum many-body systems have been the focus of several investigations, addressing their peculiar spectral and topological properties [14–16], correlation,

and entanglement patterns [17–19]. An open question concerns the long-time limit of the system under non-Hermitian dynamics, namely, whether a generalized thermalization can be expected in these systems, and if so, what are the principles underpinning this process [20, 21]. Entanglement spreading has been also discussed in the context of non-Hermitian systems, leading to a wealth of entanglement transitions in different models, due to the skin effect [22, 23] or to spectral transitions [23–30]. However, a full understanding of the structure of entanglement of non-Hermitian systems is still lacking. Integrable or exactly solvable models have played a major role in driving the understanding of (generalized) thermalization in closed systems [31] and recently in open quantum systems described by the Lindblad master equation [32, 33]. Furthermore, they have been pivotal in the understanding of entanglement dynamics in unitary and monitored systems. It is therefore natural to investigate this class of systems in the non-Hermitian case.

In this work, we focus on non-interacting, translation-invariant, fermionic non-Hermitian models, augmented with a global U(1) symmetry. We first highlight a surprising consequence of the non-linearity of non-Hermitian dynamics, namely that the presence or absence of a conserved quantity associated to a symmetry depends nontrivially on the initial state. We then argue that the interplay between symmetries, conservation laws and spectral properties of the non-Hermitian Hamiltonian allow to completely specify the structure of the steady-state at long-times. Since for Gaussian systems the entanglement entropy is fully determined by the single-particle correlation matrix, this allows also to completely characterize the entanglement content of the steady-state and the associated entanglement transitions.

In particular, we show that the steady-state of a non-Hermitian free fermionic system with complex spectrum is obtained by filling single-particle orbitals with the highest imaginary part of the eigenvalue, compatibly with the constraints imposed by the symmetries of the non-Hermitian Hamiltonian and of the initial state. As a result, the scaling of the entanglement entropy in the steady-state obeys a logarithmic law or area law depending on whether the band of imaginary eigenvalues is partially or fully filled. A striking consequence of this statement is that non-Hermitian fermionic systems support entanglement transitions, which can be tuned by the filling of the system. This feature, which has no counterpart in unitary dynamics, resembles the behavior of entanglement in the ground-states of local Hamiltonians.

For a non-Hermitian Hamiltonian with purely real spectrum, such as in the presence of $\mathcal{PT}$-symmetry [34] (or pseudo-hermiticity), we argue that the steady-state of local observables can be constructed from a diagonal ensemble similar to the one used for unitary dynamics. As a result, the entanglement entropy in the steady-state exhibits a volume-law scaling. Finally, if the spectrum is mixed with purely real and purely imaginary eigenvalues, such as in presence of $\mathcal{PT}$-symmetry breaking, the steady-state depends nontrivially on the filling and symmetries of the initial state, and so, the entanglement entropy can display either a volume-law or logarithmic law scaling.

We test our theoretical scenario on two prototypical examples of non-Hermitian lattice models: the Hatano-Nelson [35, 36] and the Su-Heeger-Schrieffer model [27]. Our results and phase diagrams for the steady-state entanglement entropy are either obtained analytically or numerically using the Faber polynomial method [37], perfectly confirming our theoretical predictions.

The manuscript is organized as follows. In Sec. 2 we introduce our general model of non-Hermitian free fermionic systems and highlight the role of the initial state in the definition of conserved quantities for non-Hermitian systems. This will lead us to introduce three classes of initial states that will be used throughout the manuscript to discuss the results. In Sec. 3 we present our main general result concerning the structure of the steady-state

occupation in our models, both in presence of complex spectrum and of $\mathcal{PT}$-symmetry, and its implication for the scaling of entanglement entropy. In Sec. 4 we present a first application of our framework to the Hatano-Nelson model and its entanglement entropy scaling in the steady-state. In Sec. 5 we discuss the non-Hermitian SSH model with its entanglement transitions and highlight the role of filling and initial state in controlling the structure of the phase diagram. Finally, Sec. 6 is devoted to the conclusions. Several appendices complete this work.

## 2   Model, Equations of Motion and Conservation Laws

In this work, we are interested in the dynamics generated by time-independent quadratic fermionic non-Hermitian Hamiltonians, that we can generally write as

$$\mathcal{H}_{\text{eff}} = \sum_{ij} \sum_{\alpha\beta} c_{i\alpha}^{\dagger} h_{ij}^{\alpha\beta} c_{j\beta}, \tag{1}$$

where $i, j$ are sites of a d-dimensional lattice, $\alpha, \beta$ additional internal quantum numbers that can include orbital, sub-lattice or spin degrees of freedom and $h_{ij}^{\alpha\beta}$ is a non-Hermitian matrix that we can write as

$$h_{ij}^{\alpha\beta} = h_{0,ij}^{\alpha\beta} + i\mathcal{V}_{ij}^{\alpha\beta} \tag{2}$$

with $h_{0,ij}^{\alpha\beta} = (h_{0,ji}^{\beta\alpha})^*$ and $\mathcal{V}_{ij}^{\alpha\beta} = (\mathcal{V}_{ji}^{\beta\alpha})^*$ respectively the real and imaginary part of the single-particle Hamiltonian. In the following, we will consider a one-dimensional lattice for simplicity, even though many of our conclusions apply more generally to non-interacting non-Hermitian fermions in higher dimensions. For our purposes, the non-Hermitianity of $\mathcal{H}_{\text{eff}}$ arises from dissipative processes such as gain and loss or more generally backaction terms associated to quantum measurements and postselection [10,38], see below for specific examples. For this reason, in the following, we will refer to the energy scale related to non-Hermiticity as dissipation or measurement backaction.

   We assume the Hamiltonian to be translational-invariant, $h_{i,j}^{\alpha\beta} \equiv h_r^{\alpha\beta}$ with $r = |i - j|$, so that we can rewrite it in momentum space as

$$\mathcal{H}_{\text{eff}} = \sum_{k} \sum_{\alpha\beta} c_{k\alpha}^{\dagger} h_k^{\alpha\beta} c_{k\beta}, \tag{3}$$

with $h_k^{\alpha\beta} = \sum_r e^{ikr} h_r^{\alpha\beta}$ the single-particle Hamiltonian in momentum space. This Hamiltonian can be decomposed into its Hermitian part, $h_{0,k}$, which is responsible for the unitary evolution in the absence of dissipative processes, and the dissipative contribution, $i\mathcal{V}_k$,

$$h_k^{\alpha\beta} = h_{0,k}^{\alpha\beta} + i\mathcal{V}_k^{\alpha\beta}, \tag{4}$$

with $h_{0,k} = h_{0,k}^{\dagger}$. We assume that this Hamiltonian can be diagonalised with eigenvalues $\varepsilon_{k,a} = \text{Re}\,\varepsilon_{k,a} + i\,\text{Im}\,\varepsilon_{k,a}$. This describes the complex energies of the non-Hermitian quasiparticles of the problem which have a finite decay rate $\gamma_{k,a} = \text{Im}\varepsilon_{k,a}$.

   The system evolution under $\mathcal{H}_{\text{eff}}$ reads

$$|\Psi(t)\rangle = \frac{e^{-i\mathcal{H}_{\text{eff}}t}|\Psi(0)\rangle}{\|e^{-i\mathcal{H}_{\text{eff}}t}|\Psi(0)\rangle\|}, \tag{5}$$

where the denominator properly normalizes the state, compensating the non-conservation of the state's norm due to the absence of unitarity in non-Hermitian systems. The equation

above can be obtained as a no-click limit of the quantum jump dynamics of monitored systems [6]. Normalizing the state after the evolution is equivalent to solving a non-linear Schrödinger equation

$$d|\Psi(t)\rangle = -i\mathcal{H}_{\text{eff}}dt|\Psi(t)\rangle - i\frac{dt}{2}\langle\mathcal{H}_{\text{eff}} - \mathcal{H}_{\text{eff}}^{\dagger}\rangle_t|\Psi(t)\rangle. \tag{6}$$

where the last term, proportional to the imaginary part of the Hamiltonian, depends non-linearly on the state itself.

In this manuscript, we focus on the dynamics described by Eq. (5),(6) and discuss the steady-state properties of the system and its entanglement content. The non-Hermitian nature of the problem has direct consequences on the Heisenberg equations of motion of any operator, $\mathcal{O}$, evolving under $\mathcal{H}_{\text{eff}}$. Indeed, the equation of motion is given by

$$\partial_t \langle\mathcal{O}\rangle = i\left\langle\mathcal{H}_{\text{eff}}^{\dagger}\mathcal{O} - \mathcal{O}\mathcal{H}_{\text{eff}}\right\rangle - i\left\langle\mathcal{H}_{\text{eff}}^{\dagger} - \mathcal{H}_{\text{eff}}\right\rangle\langle\mathcal{O}\rangle. \tag{7}$$

The last term arises explicitly from the measurement-back action (i.e. from the non-Hermitian part of the Hamiltonian) and vanishes in Hermitian systems, thereby recovering the standard Heisenberg equation of motion. This term induces a non-linear, state dependent form to the equations of motion, even for a quadratic non-interacting systems such as in Eq.(1). To appreciate this point, it is useful to write down explicitly the equations of motion for the correlation matrix $\mathcal{G}_{ij\alpha\beta} = \langle c_{i\alpha}^{\dagger}c_{j\beta}\rangle$, which for a fermionic gaussian problem whose dynamics conserves particle number encodes all relevant information. In momentum space $G_{kq}^{\alpha\beta} = \langle c_{k\alpha}^{\dagger}c_{q\beta}\rangle$, one can show using Wick's theorem and the decomposition in Eq. (4) that the diagonal part satisfies the following equation of motion,

$$\partial_t G_{kk}^{\alpha\beta} = i\left((h_k^*)^{\alpha\gamma} G_{kk}^{\gamma\beta} - G_{kk}^{\alpha\gamma}h_k^{\beta\gamma}\right) - 2\sum_q G_{kq}^{\alpha\xi} (\mathcal{V}_q^*)^{\xi\gamma} G_{qk}^{\gamma\beta}, \tag{8}$$

where repeated indices are summed. We note that the equation is non-linear due to the non-Hermitian part of the Hamiltonian. Furthermore, even for a translation-invariant Hamiltonian, the diagonal elements of the correlation matrix are coupled to the off-diagonal ones due to the non-Hermitian term in the Hamiltonian. This is not the case for a Hermitian system, as the last term is absent. This already suggests that symmetries and conserved quantities play a special role for non-Hermitian systems. Finally, for what concerns the entanglement content of the steady-state, we characterize it using the von Neumann entanglement entropy, defined as [39, 40]

$$S(t) = -\text{tr}_A\left[\rho^A(t)\ln\rho^A(t)\right], \tag{9}$$

where we have introduced a bipartition $A \cup B$ in the system, with the reduced density matrix $\rho^A(t) = \text{tr}_B|\Psi(t)\rangle\langle\Psi(t)|$. The von Neumann entropy correctly measures entanglement, as the system is initially in a pure state and purity is conserved throughout the time evolution, as seen by Eq. (5). As we focus on quadratic models, the entanglement entropy can be directly obtained from the correlation matrix [41].

## 2.1 Impact of the Initial State on Conservation Laws

We now discuss the definition of a conserved quantity for non-Hermitian dynamics and the influence of the initial state, which play a crucial role in the dynamical evolution.

In a Hermitian system with Hamiltonian $\mathcal{H}$ a time-independent observable $\mathcal{Q}$ is conserved if $[\mathcal{H}, \mathcal{Q}] = 0$, as directly given by the Heisenberg equation of motion: $\partial_t \langle\mathcal{Q}\rangle = i[\mathcal{H}, \mathcal{Q}]$.

In this case, the expectation value of $\mathcal{Q}$ is fixed by the initial state, $\langle \Psi_0 | \mathcal{Q} | \Psi_0 \rangle$. All symmetries of the Hamiltonian correspond to a conserved quantity, and vice versa, as encapsulated by the Noether theorem, irrespectively of whether the initial state respects the symmetry.

Non-Hermitian systems, on the other hand, exhibit peculiar behavior due to their non-linear state-dependent evolution equation. In particular, the conservation laws in non-Hermitian systems are not solely determined by the symmetries of the Hamiltonian. As in other settings of open quantum systems, such as the Lindblad master equation, the existence of a (weak) symmetry in the dynamics does not imply the presence of a conserved quantity [6]. For a non-Hermitian system, in addition to the symmetries, the initial state also plays a key role in determining the existence or not of a conserved quantity. Indeed, as one can see from Eq. (7) for non-Hermitian systems in order for the expectation value of an operator $\mathcal{Q}$ to be conserved by the non-unitary evolution, two conditions are required, namely: (i) the operator has to commute with $\mathcal{H}_{\text{eff}}$, i.e., $[\mathcal{H}_{\text{eff}}, \mathcal{Q}] = 0$ and (ii) the initial state must be an eigenstate of the operator $\mathcal{Q}$, i.e. $\mathcal{Q}|\Psi_0\rangle = q_0|\Psi_0\rangle$. These two conditions together guarantee that the state remains an eigenstate of the conserved quantity throughout the evolution. In fact, the equation of motion is identically zero because the following identity holds $\left\langle \mathcal{H}_{\text{eff}}^{\dagger} \mathcal{Q} - \mathcal{Q} \mathcal{H}_{\text{eff}} \right\rangle = \left\langle \mathcal{H}_{\text{eff}}^{\dagger} - \mathcal{H}_{\text{eff}} \right\rangle \langle \mathcal{Q} \rangle$. In other words, we can write

$$[\mathcal{H}_{\text{eff}}, \mathcal{Q}] = 0, \quad \text{and} \quad \mathcal{Q}|\Psi_0\rangle = q_0|\Psi_0\rangle \Rightarrow \partial_t \langle \mathcal{Q} \rangle = 0. \tag{10}$$

To appreciate the role of the initial state in determining the conservation laws of our system, it is useful to note that our non-Hermitian Hamiltonian in Eq.(1) admits both a U(1) symmetry, $c_{i\alpha} \to e^{i\varphi} c_{i\alpha}$ and a symmetry under spatial translation. As a result, we can identify two operators that commute with $H_{\text{eff}}$: the number of fermions at momentum $k$,

$$\hat{n}_k = \sum_\alpha c_{k,\alpha}^{\dagger} c_{k,\alpha}, \tag{11}$$

as well as, the total number of particles,

$$\hat{N} = \sum_k \hat{n}_k. \tag{12}$$

We note that, conversely, the occupation of the eigenmodes of $\mathcal{H}_{\text{eff}}$ are not in general Hermitian operators and, therefore, not of direct utility for our purpose here. To assess whether to these operators we can associate genuine conserved quantities, we will have to consider three distinct classes of initial states $|\Psi_0\rangle$.

- class **A**: $|\Psi_0\rangle$ is an eigenstate of $\hat{n}_k$, for all $k$. In general, this condition gives a Slater determinant of the form

$$|\Psi_0\rangle = \prod_{k \in \text{occupied}} \eta_k^{\dagger} |\text{vac}\rangle, \tag{13}$$

  with $\eta_k = \sum_\beta U_{k,\beta} c_{k,\beta}$ with $U_k$ satisfying $\sum_\beta |U_{k,\beta}|^2 = 1$. For initial states in this class, the expectation value of $\hat{n}_k$ is conserved even under the non-Hermitian evolution. In other words, states in this class conserve the total particle number and are translationally invariant.

- class **B**: $|\Psi_0\rangle$ is an eigenstate of $\hat{N}$ but not an eigenstate of $\hat{n}_k$, hence the expectation value of $\hat{n}_k$ admits a nontrivial dynamics, while the total particle number is a genuine

conserved quantity of the non-Hermitian dynamics. An example of initial state in this class, that we will use later on in the manuscript, read:

$$|\Psi_0\rangle = \prod_{l=0}^{N-1} c_{\sigma(l)}^\dagger |\text{vac}\rangle, \tag{14}$$

where $\sigma$ is a permutation of $\{1, ..., L\}$ with $L$ the size of the chain.

- class **C**: $|\Psi_0\rangle$ is neither an eigenstate of $\hat{n}_k$ nor of the particle number $\hat{N}$, therefore, the associated expectation values are not conserved during the dynamics. Typically, such a state can be an eigenstate of Bogoliubov-de-Gennes Hamiltonians and can be written in the form

$$|\Psi_0\rangle = \prod_{l=0}^{L/2-1} \left[ u_l + v_l c_{2l}^\dagger c_{2l+1}^\dagger \right] |\text{vac}\rangle, \tag{15}$$

where $|u_l|^2 + |v_l|^2 = 1$ to assure the correct normalization. One useful quantity is then the average number of particles, $\left\langle \hat{N} \right\rangle$, and the fluctuations of the total particle number $\left\langle \delta \hat{N}^2 \right\rangle = \left\langle \hat{N}^2 \right\rangle - \left\langle \hat{N} \right\rangle^2$, which for the initial state written above are given by,

$$\left\langle \hat{N} \right\rangle = 2 \sum_{l=0}^{L/2-1} |v_l|^2, \quad \left\langle \delta \hat{N}^2 \right\rangle = 4 \sum_{l=0}^{L/2-1} |u_l|^2 |v_l|^2, \tag{16}$$

respectively.

In the following, we will discuss the general features of the long-time dynamics of non-Hermitian free fermionic systems, starting from initial states in these three classes.

# 3 Long-time limit, Steady-State Occupation and Entanglement Entropy

We now present our main result concerning the long-time limit of a free non-Hermitian fermionic system and its entanglement structure, which we will then discuss in detail for specific examples in the remainder of the manuscript (see Sections 4 and 5). Specifically, here we aim at characterizing the steady-state of the system in terms of the occupation of single-particle states $\langle \hat{n}_k \rangle$, whose dynamics as discussed above depends strongly on the initial condition. This will allow us to understand the role that the filling of the system can have and the consequences we can observe on the scaling of entanglement entropy.

## 3.1 Steady-state structure: Complex Spectrum

To make general statements about the steady-state structure, we assume the non-Hermitian Hamiltonian to be diagonalizable in a bi-orthogonal basis,

$$\mathcal{H}_{\text{eff}} = \sum_n (E_n + i\Gamma_n) \left| \Psi_n^R \right\rangle \left\langle \Psi_n^L \right|, \tag{17}$$

where $\left| \Psi_n^{R/L} \right\rangle$ is the many-body right/left $n^{th}$ eigenstate and $(E_n + i\Gamma_n)$ the associated eigenvalue that we take here to be complex, i.e. we will assume $\Gamma_n$ to be non-zero for any $n$. In general, $\Gamma_n$ can be either positive (describing growing modes) or negative (describing

decaying or damped modes). However, due to the normalization in the non-Hermitian evolution, the spectrum is defined up to a global shift, so its overall sign is irrelevant: only differences among eigenvalues matter.

The evolution of the state (given in Eq. (5)) can be written using this bi-orthogonal basis by inserting the identity $\mathbf{1} = \sum_n \left|\Psi_n^R\right\rangle\left\langle\Psi_n^L\right|$ between the initial state and the time evolution operator,

$$|\Psi(t)\rangle = \frac{e^{\Gamma_{\rm ss}t}}{\mathcal{N}(t)}\left(e^{-iE_{\rm ss}t}\langle\Psi_{\rm ss}^L|\Psi(0)\rangle\,|\Psi_{\rm ss}^R\rangle + \sum_{n\neq{\rm ss}}e^{(\Gamma_n-\Gamma_{\rm ss})t}e^{-iE_nt}\langle\Psi_n^L|\Psi(0)\rangle\,|\Psi_n^R\rangle\right), \quad (18)$$

where $\mathcal{N}(t)$ properly normalizes the state, and we identified with the index $n = {\rm ss}$ the pair of right and left eigenstates $\left|\Psi_n^{R/L}\right\rangle$ that have the highest $\Gamma_n$, the imaginary part of the eigenvalue, among those with a non-zero overlap with the initial state, $\langle\Psi_{\rm ss}^L|\Psi(0)\rangle \neq 0$. For simplicity, we have assumed that the state with the highest $\Gamma_n$ is non-degenerate. In the long time limit, we obtain

$$\lim_{t\to+\infty}|\Psi(t)\rangle = e^{-iE_{\rm ss}t}|\Psi_{\rm ss}^R\rangle. \quad (19)$$

The interpretation of this statement is simple: the state which is populated at long times is the one with highest imaginary part of the eigenvalue, corresponding to the fastest growth rate (if $\Gamma_{\rm ss} > 0$) or slowest decay rate (if $\Gamma_{\rm ss} < 0$).

We can say more about the structure of the steady-state by taking into account the symmetries of the non-Hermitian Hamiltonian. Indeed, both the left and right eigenvectors are eigenstates of the Hamiltonian's symmetries. For the non-Hermitian quadratic Hamiltonian considered here, this implies that each eigenstate is labeled by the occupation number, $n_k$, for each momentum $k$, and the total particle number, $N$, namely we have that

$$\begin{aligned}\forall\,k,\ \hat{n}_k\left|\Psi_m^{R/L}\right\rangle &= (n_k)_m\left|\Psi_m^{R/L}\right\rangle,\\ \hat{N}\left|\Psi_m^{R/L}\right\rangle &= N_m\left|\Psi_m^{R/L}\right\rangle,\end{aligned} \quad (20)$$

where $(n_k)_m$ and $N_m$ are the eigenvalues corresponding to $m^{\rm th}$ eigenstate. Finally, the detailed structure of the steady-state differs for the various classes of initial states considered in this work, as we now discuss.

- An initial state in class **A** is also an eigenstate of the occupation number $\hat{n}_k$, for each momentum $k$ with $\hat{n}_k|\Psi_0\rangle = (n_k)_{t=0}|\Psi_0\rangle$, and the total particle number, $\hat{N}$ with $\hat{N}|\Psi_0\rangle = N_{t=0}|\Psi_0\rangle$. Consequently, as imposed by Eq. (7), throughout the dynamics, the average occupation for each momentum, $\langle\hat{n}_k\rangle$ remains conserved and fixed at its initial value, $(n_k)_{t=0}$. As a result, the steady-state in this case corresponds to the right eigenstate of the Hamiltonian characterized by $(n_k)_{t=0}$ quantum numbers, with the highest imaginary eigenvalue, $\Gamma_{\rm ss}$. This component can be written using the single-particle energies as follows

$$\Gamma_{\rm ss} = \sum_k \sum_{\alpha=0}^{(n_k)_{t=0}-1} {\rm Im}\,\varepsilon_{k,\alpha}, \quad (21)$$

  where $(n_k)_{t=0}$ is the eigenvalue of the $\hat{n}_k$ operator in the initial state, $\alpha$ corresponds to a band index of the single-particle spectra, and we have assumed the ordering convention ${\rm Im}\,\varepsilon_{k,0} > {\rm Im}\,\varepsilon_{k,1} > \cdots$. This is represented in the scheme of Fig. 1.

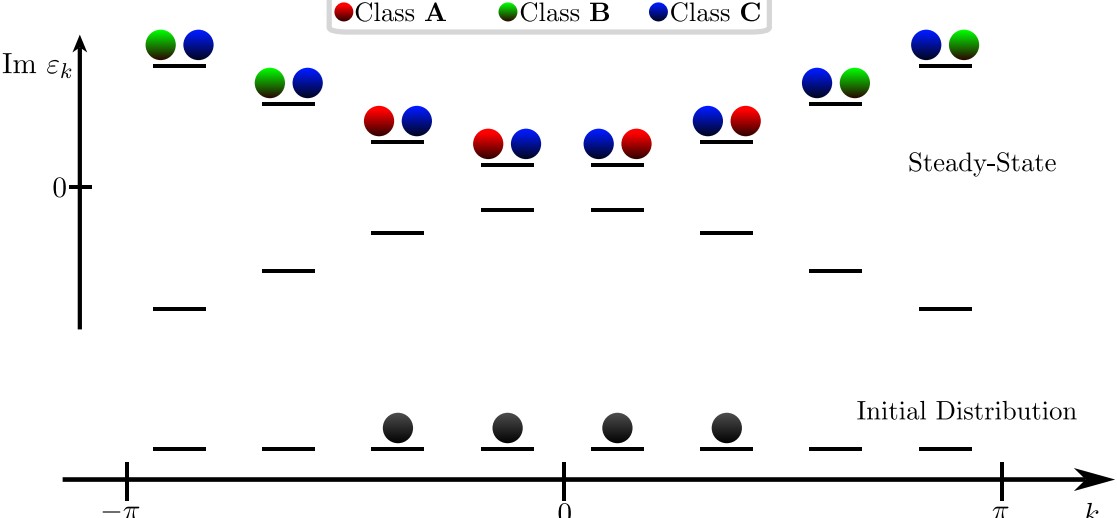

**Figure 1:** Sketch of the steady-state occupation of single-particle energy levels by class. For simplicity, we consider the case of a two-band single-particle Hamiltonian. The black balls represent the initial distribution. This should be interpreted as the expectation value of $\hat{n}_k$ for class **B** and class **C**.

The single-particle states with the largest imaginary part of the eigenvalue are filled, preserving the total particle number and initial momentum occupation. As seen in the example of Fig. 1, the number of red balls matches the number of black ones, and both occupy the same momentum values.

- For an initial state in class **B** the situation is distinct. In this case, the initial state only has a well-defined particle number. The expectation value $\langle \hat{n}_k \rangle$ is no longer conserved under the evolution, as the initial state as a non-zero overlap with eigenstates that have different values of $(n_k)_m$. In this case, the steady state is the right eigenstate that maximizes the imaginary part of the eigenvalue, while maintaining the same particle number as the initial state, $\langle \hat{N} \rangle_{t=0}$.

The steady state is then obtained by filling the single-particle states with the largest imaginary parts of the eigenvalue, subject to the total particle number constraint. Given this,

$$\Gamma_{\text{ss}} = \sum_{m=0}^{\langle \hat{N} \rangle_{t=0}-1} \text{Im}\ \varepsilon_m, \tag{22}$$

where $\langle N \rangle_0$ is the eigenvalue of total particle number operator correspondent to the initial state, $m = (k, \alpha)$ and we assume the ordering convention $\text{Im}\ \varepsilon_0 > \text{Im}\ \varepsilon_1 > \cdots$. This last equation also tells us the value of the distribution $(n_k)_{\text{ss}}$ in the steady-state. As seen in the scheme of Fig. 1, the number of occupied final states (depicted in green) matches the number of initial occupied states (represented in black), but they do not occupy the same momentum values.

- In class **C**, the initial states are neither eigenstates of the occupation in momentum nor of the total number operator. Generically, the initial state overlaps with all particle number sectors. Thus, in general, the steady state of the system corresponds to the right eigenstate with the highest imaginary part of the eigenvalue in the full many-body spectrum. This corresponds to filling all single-particle states that have

a positive imaginary eigenvalue, corresponding to amplifying modes

$$\Gamma_{ss} = \sum_m \operatorname{Im} \varepsilon_m, \quad \operatorname{Im} \varepsilon_m \geq 0, \tag{23}$$

where $m = (k, \alpha)$ and we assume that the imaginary part of the single-particle spectrum has a zero center of mass, meaning the imaginary part of the sum of all single-particle eigenvalues is zero. In the example of Fig. 1, the number of blue balls does not match that of black ones.

We can summarize these considerations and express the steady-state occupation of single-particle modes $n_k$ as

$$(n_k)_{ss} = \begin{cases} (n_k)_{t=0}, & \text{class } \mathbf{A}, \\ \Theta(\operatorname{Im} \varepsilon_{k,\alpha} - \mu_{\text{eff}}), & \text{class } \mathbf{B}, \\ \Theta(\operatorname{Im} \varepsilon_{k,\alpha}), & \text{class } \mathbf{C}, \end{cases} \tag{24}$$

where $(n_k)_{ss}$ is the eigenvalue of the operator $\hat{n}_k$ in the steady state, $\Theta(\cdot)$ denotes the Heaviside step function, and $\mu_{\text{eff}}$ is an effective chemical potential ensuring that the total particle number remains equal to that of the initial state, i.e., $\sum_k (n_k)_{ss} = N_0$. As before, we assume that the imaginary part of the single-particle spectrum has a zero center of mass.

We note that the steady state given by Eq. 18 possesses the same symmetries as the considered non-Hermitian Hamiltonian, although the initial state did not. Interestingly, this implies that time evolution allows for the restoration of the symmetries, even though they were initially explicitly broken by the state. For example, an initial state in class **C** is a coherent superposition of states with distinct values of the total particle number, but the steady state is an eigenstate of the total particle number. We will return to this point when discussing specific examples.

We conclude by noticing the formal analogy between what we discussed so far for non-Hermitian evolution concerning filling of modes with highest imaginary eigenvalue, and the long-time limit of a state evolved in imaginary time with a Hermitian Hamiltonian. In this case, the initial state converges to the ground-state of the Hamiltonian within the sector that has the same quantum numbers as the initial state and has the lowest (real) energy. This can also be viewed as real-time evolution under the non-Hermitian Hamiltonian $\mathcal{H}_{\text{eff}} = -iH$, where $H$ is a Hermitian operator. Thus, minimizing the real energy in imaginary-time evolution corresponds to maximizing the imaginary energy in real-time evolution.

## 3.2 $\mathcal{PT}$ Symmetric Hamiltonians

In the previous section we have assumed a complex energy spectrum. However, in non-Hermitian physics an important class of problems features $\mathcal{PT}$-symmetry [34]. In this case, the non-Hermitian Hamiltonian $H_{\text{eff}}$ commutes with the $\mathcal{PT}$ operator, the conjugation of the parity ($\mathcal{P}$) and time-reversal ($\mathcal{T}$) operators, that is, $[\mathcal{H}_{\text{eff}}, \mathcal{PT}] = 0$. This symmetry ensures that the spectrum of the Hamiltonian is real whenever the eigenstates are also eigenvectors of the $\mathcal{PT}$ operator [34]. Depending on the values of the system's parameters, this condition may or may not hold. When it does, the system is said to be in a $\mathcal{PT}$-symmetric phase. Conversely, if the condition is not satisfied, the spectrum becomes complex and the system is in a $\mathcal{PT}$-broken phase, as the state spontaneously breaks the $\mathcal{PT}$ symmetry. The breaking of $\mathcal{PT}$ symmetry can also occur in stages, with only part of

the spectrum developing imaginary eigenvalues, before the full spectrum becomes complex, as we will discuss in an example later in the manuscript.

In the $\mathcal{PT}$-symmetric phase, the steady-state cannot be described by Eq. (18) as all eigenstates are real, and so, in principle, none is a priori more amplified/damped than the others. Although the time evolution is still non-unitary due to the non-orthogonality between distinct right/left eigenstates, $\langle \Psi_n^R | \Psi_m^R \rangle \neq 0$, we can understand the structure of the steady-state using an analogy with the emergence of the diagonal ensemble under unitary dynamics in closed isolated quantum systems. As we show in the appendix B, the expectation value of local observables in the steady-state can be approximated by a diagonal ensemble of the form:

$$\rho_{\mathrm{DE}} = \frac{1}{\mathcal{Z}_D} \sum_n \left| \langle \Psi(0) | \Psi_n^L \rangle \right|^2 \left| \Psi_n^R \right\rangle \left\langle \Psi_n^R \right|, \tag{25}$$

where $\mathcal{Z}_{\mathrm{DE}} = \sum_n \left| \langle \Psi(0) | \Psi_n^L \rangle \right|^2$ is the normalization factor. We note that the diagonal ensemble explicitly depends on the initial state of the system. The density matrix of Eq. (25) is constructed only with the right eigenstates that have the same quantum numbers as the initial state, since otherwise the overlap $\langle \Psi(0) | \Psi_n^L \rangle$ is zero. We stress that, as its unitary case counterpart, Eq. (25) should only be understood as valid when evaluating the long-time expectation value of sufficiently local operators. The full wave function remains in a pure state under non-Hermitian evolution, yet local observables can reach a steady-state via many-body dephasing (see Appendix B). Finally, we mention that a similar construction of the diagonal ensemble could be useful for other types of pseudo-Hermitian Hamiltonians (among which $\mathcal{PT}-$ symmetric Hamiltonians represent one possible example).

If instead the system breaks spontaneously $\mathcal{PT}$ symmetry and develops imaginary eigenvalues (i.e. in the $\mathcal{PT}$-broken phase), the structure of the steady-state can be still understood using the argument in previous section, at least when all the spectrum becomes complex. In this scenario, the steady state corresponds to the least damped mode subject to the activated conservation laws. We emphasize that for certain values of the Hamiltonian parameters, exceptional points may arise, preventing us from diagonalizing the full Hamiltonian. In this case, the construction of Eq. (18) cannot be made. Typically, these exceptional points appear in the spectrum along with spontaneous symmetry breaking of $\mathcal{PT}$ symmetry [24]. We discuss this pathological case later in the manuscript.

## 3.3 Entanglement Entropy

We now use the knowledge of the steady-state occupation discussed above to understand the structure of the entanglement entropy in the steady state, $S(+\infty) = \lim_{t \to +\infty} S(t)$, where $S(t)$ is defined in Eq. (9). As in previous sections, we focus here on one-dimensional fermions, although our conclusions can be easily extended to higher dimensions. Moreover, since we are considering Gaussian fermionic states, knowledge of the single-particle correlation matrix is sufficient to obtain the entanglement entropy, at least numerically. Remarkably, we argue that one can obtain exact analytical predictions for the scaling of the entanglement entropy by knowing the steady-state occupation. Quite generally we can write the steady-state entanglement entropy of a bi-partition of linear size $\ell$ in the form

$$S(+\infty) = a_1 \ell + a_2 \ln \ell + \mathcal{O}(1), \tag{26}$$

The coefficient $a_1$ corresponds to the extensive volume-law contribution to the entanglement entropy, $a_2$ controls the sub-extensive term characteristic of critical behavior, and finally we have omitted the constant term corresponding to the area-law scaling. While

Eq. (26) encompasses all cases of interest for our models, we show in Sec. 5 and Appendix C how the coefficients $a_1, a_2$ can be computed analytically in certain cases.

To see how Eq. (26) comes about, let us start by discussing the case when the spectrum is complex and our discussion in Sec. 3.1 applies (see Eq. (18)). If this is the case, then the entanglement entropy in the steady state is fully determined by the distribution $\langle \hat{n}_k \rangle$ and the structure of the imaginary spectrum of the single-particle Hamiltonian. In this regime, the steady-state correlation matrix is diagonal in the momentum index, that is, $G_{kq}^{\alpha\beta} = 0$ for $k \neq q$. For a given distribution of $\langle \hat{n}_k \rangle$ and an imaginary single-particle spectrum, the steady state resembles a Slater-determinant characteristic of fermions with a partially or completely filled band of imaginary eigenvalues. Consequently, if the eigenstates of the non-Hermitian Hamiltonian can be analytically determined, the steady-state entanglement entropy can also be explicitly obtained for all classes of initial states using the Szegö theorem [3] and the Fisher-Hartwig conjecture [42]. When the steady state exhibits a fully occupied band of imaginary eigenvalues, and the final occupied band is separated by an energy gap from the next band in the imaginary part of the spectrum, the entanglement entropy obeys an area law. In contrast, when the spectrum is gapless, it exhibits a critical sub-volume logarithmic scaling. Generically, if it exhibits metallic-like behavior, i.e. a partially filled band, the system's entanglement entropy will grow logarithmically with subsystem size.

This behavior mirrors the scaling of the entanglement entropy observed in the ground-states of models with short-range interactions and hopping [43, 44]. However, for the quadratic non-Hermitian Hamiltonians considered here, the structure of the imaginary part of the single-particle spectrum (i.e. whether it is gapped or gapless) alone does not determine the entanglement entropy scaling. Instead, the momentum-space occupation, which depends on the initial state and on the initial filling, also plays a crucial role.

Let us now consider the case of a non-Hermitian Hamiltonian with $\mathcal{PT}$ symmetry. If this symmetry is unbroken, all eigenstates share the same imaginary eigenvalue (zero), leading to a high degree of degeneracy. In these cases, Eq. (18) is no longer applicable. Instead, the steady state is typically a linear combination of the Hamiltonian's eigenstates, and the correlation matrix is generically not diagonal in momentum space. As a result, the Szegö theorem and the Fisher-Hartwig conjecture cannot be directly used. Based on our discussion of the steady-state occupation and the emergence of a diagonal ensemble, we can expect the entanglement entropy to satisfy a volume-law scaling, in analogy with closed systems. This can be checked explicitly if the initial state belongs to class **A**, when the correlation matrix remains diagonal in momentum space. In this case, the computation is feasible, provided that the equations of motion are solved exactly and the steady-state correlation matrix is determined, as we will show in Sec. 5 for the SSH model.

Finally, if the spectrum contains both purely real and purely imaginary eigenvalues, such as when the $\mathcal{PT}$ symmetry is broken, the entanglement entropy depends on the filling of the imaginary and real eigenstates, with possible transition between volume-law and subvolume (logarithmic) scaling. We will discuss a specific example of this case in Sec. 5.

In the table of Fig. 2, we present a summary of the possible steady-state entanglement scalings as a function of the distribution of single-particle states in the steady state. As we have explained, the specific distribution depends on both the spectrum of the non-Hermitian Hamiltonian and the conservation laws imposed by the initial state.

| Spectrum | Steady State | Purely Real Energy Eigenstates | Complex Energy Band | Band Occupation | Entanglement scaling |
|---|---|---|---|---|---|
| Complex | $\lvert \Psi_{\mathrm{ss}}^{R} \rangle$ | N.A. | Fully filled | | $S \propto \mathcal{O}(1)$ |
| | | N.A. | Partially filled | | $S \propto \ln \ell$ |
| | | N.A. | Fully filled | | $S \propto \ln \ell$ |
| Mixed | | Empty | Partially filled | | $S \propto \ln \ell$ |
| | | Occupied | Partially filled | | $S \propto \ell$ |
| | | Fully Occupied | Partially filled | | $S \propto \ln \ell$ |
| Purely Real | $\rho_{\mathrm{DE}}$ | Occupied | N.A. | | $S \propto \ell$ |

○ Empty mode    ● Filled Mode    ● Occupied Purely Real Mode    ● Fully Occupied Purely Real Mode

**Figure 2:** Scaling of steady-state entanglement entropy, depending on the spectrum of the non-hermitian Hamiltonian and the structure of the steady state. In the *band occupation* column, we illustrate the case of a two-band model for simplicity.

## 4   Application: Hatano-Nelson Model

We start by analyzing the paradigmatic Hatano-Nelson model [35, 36] with periodic boundary conditions, a one-dimensional chain of spinless fermions with an asymmetric hopping, whose real-space Hamiltonian is given by

$$\mathcal{H} = -\frac{1}{2} \sum_{l=0}^{L-1} \left( [J + \gamma]\, c_l^\dagger c_{l+1} + [J - \gamma]\, c_{l+1}^\dagger c_l \right) = \sum_k \varepsilon_k c_k^\dagger c_k, \qquad (27)$$

where $J$ is the coherent hopping strength to the first-neighbor sites, $\gamma \in \mathbb{R}$ induces an imbalance in the charge hopping and $c_l^\dagger$ ($c_l$) is the creation (annihilation) operator which creates (destroys) a fermion on site $l$. The Hatano-Nelson Hamiltonian is diagonalized by Fourier transforming the fermionic operators, and as such the Hamiltonian is diagonal in the momentum occupation $\hat{n}_k = c_k^\dagger c_k$ with a complex single-particle dispersion relation

$$\varepsilon_k = -J \cos(ka) + i\gamma \sin(ka). \qquad (28)$$

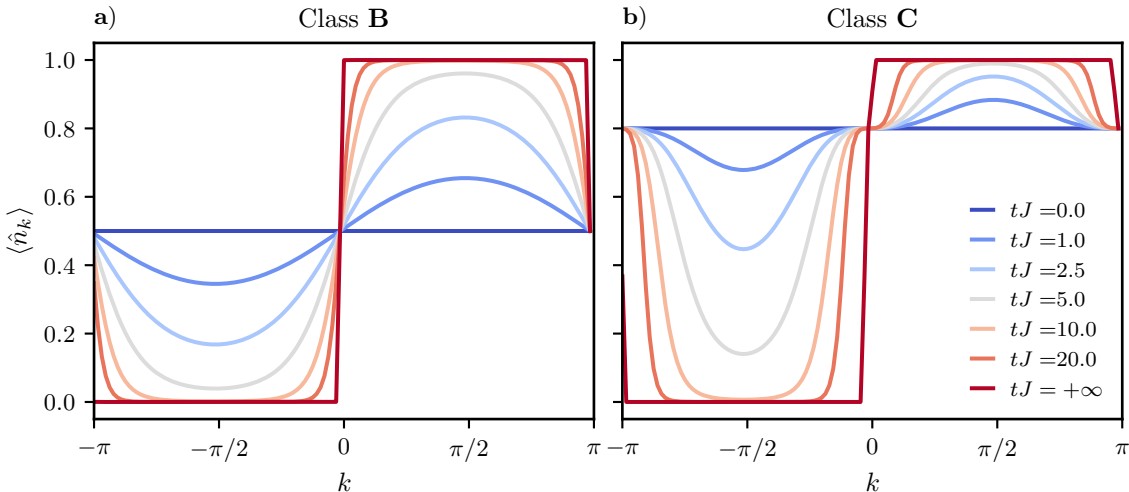

**Figure 3:** Time evolution of the expectation value of $\langle \hat{n}_k \rangle$ distribution. In panel a), the initial state corresponds to a charge density wave, $|\Psi_0\rangle = |101 \cdots 101\rangle$, and so it is in class **B** (corresponding to an initial filling equal to 1/2), while in panel b) the initial state belongs to class **C**. Other parameters: $L = 128$ and $\gamma = 0.4J$.

We see that taking $\gamma > 0$, the imaginary part of the spectrum is positive for $0 < k < \pi$ and negative otherwise.

We start considering an initial state in class **A**, corresponding to

$$|\Psi_0\rangle = \prod_{k \in \Omega} c_k^\dagger |\text{vac}\rangle , \qquad (29)$$

where $\Omega$ a collection of the values of momenta $k$. For this specific case, the time evolution is trivial, as the initial state is an eigenstate of the Hamiltonian, and so the state does not evolve in time

$$|\psi(t)\rangle = e^{-i\text{Re}(E_k)t} |\psi_0\rangle , \qquad (30)$$

where $E_k = \sum_{k \in \Omega} \varepsilon_k$. As expected, the state maintains the expectation value of $\langle \hat{n}_k \rangle$ for all $k$. Moreover, the entanglement entropy in the steady state is exactly the same as in the initial state.

On the other hand, the state evolves nontrivially for an initial state in either class **B** or C. Using the considerations discussed in the previous section and Eq. (18), the steady state for an initial state in class **B** is a Slater determinant obtained by filling the single-particle eigenstates with the largest imaginary part. For the single-particle dispersion relation in Eq. (28) and for $\gamma > 0$ this corresponds to filling the single-particle eigenstates around $k = \pi/2$,

$$|\Psi_{\text{ss}}^B\rangle = \prod_{k \in \left[\frac{\pi}{2} - k_f, \frac{\pi}{2} + k_f\right]} c_k^\dagger |\text{vac}\rangle , \qquad (31)$$

where $k_f$ is determined by the total particle number of the initial state.

With this in mind, the distribution is given by

$$(n_k)_{\text{ss}} = \begin{cases} 1, & k \in \left[\frac{\pi}{2} - k_f, \frac{\pi}{2} + k_f\right] , \\ 0, & \text{otherwise.} \end{cases} \qquad (32)$$

We show this in the panel a) of Fig. 3, where we plot the dynamics of the occupation number $\langle \hat{n}_k \rangle$ for increasing values of time. We see that for an initial state at half filling,

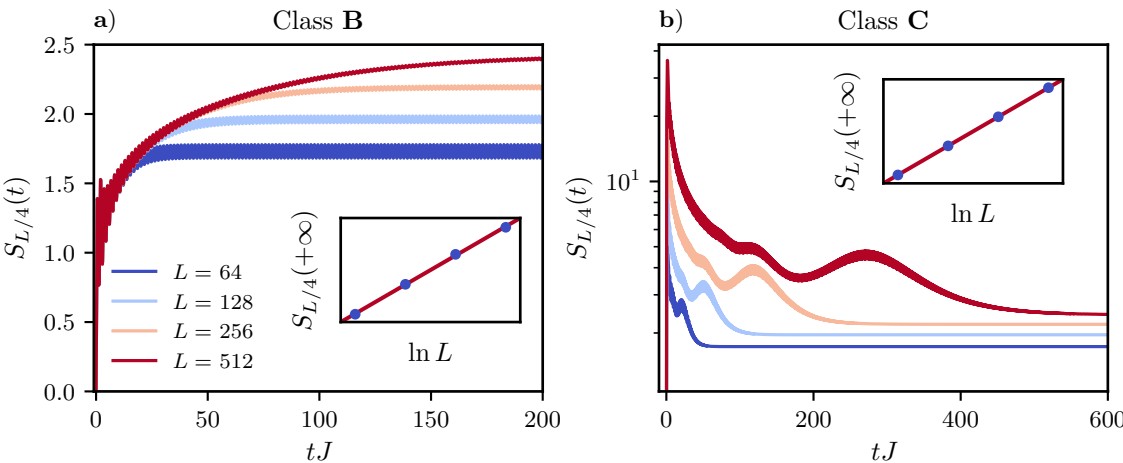

**Figure 4:** Time evolution of the entanglement entropy for different system sizes, starting from initial states in class **B** (panel a)) and class **C** (panel b)). The inset in both panel shows the steady-state entanglement entropy as a function of the total system size. The scaling behavior is found to be $S(+\infty) \propto a_2 \ln(L) + \mathcal{O}(1)$, with $a_2 = 0.321 \pm 0.05$ in panel a and $a_2 = 0.3336 \pm 0.0005$ in panel b). This is consistent with the analytical prediction $a_2 = 1/3$ for the thermodynamic limit. Other parameters: $\gamma = 0.4J$.

$\langle \hat{n}_k(0) \rangle = 1/2$, the non-Hermitian dynamics fills up the states around $k = \pi/2$ in the long-time limit, while depleting states with negative imaginary part of the eigenvalues, corresponding to decaying modes, in such a way that the total number of particles remains preserved. For an initial state in class **C**, based on our general framework, we expect the steady state to be of the form

$$|\Psi_{ss}^C\rangle = \prod_{k \in [0,\pi]} c_k^\dagger |vac\rangle, \tag{33}$$

corresponding to a Slater determinant with the highest imaginary eigenvalue. Since for initial states in class **C** there is no particle number conservation, this steady state corresponds to filling all single-particle states with positive energy. For $\gamma > 0$, this corresponds to fill $k \in [0, \pi]$, thus

$$(n_k)_{ss} = \begin{cases} 1, & k \in [0, \pi], \\ 0, & \text{otherwise.} \end{cases} \tag{34}$$

We observe this in the numerical results shown in the panel b) of Fig. 3. The expectation value of $\langle \hat{n}_k \rangle$ evolves until all states with a positive imaginary energy are occupied. In contrast with the state in class **B** the area under the curve, corresponding to the total particle number, is not conserved throughout evolution.

Given the structure of the steady-state occupation, we can immediately infer the steady-state entanglement entropy for the Hatano-Nelson model, which can also be computed analytically by using the Fisher-Hartwig conjecture (see the Appendix C). Indeed, for both classes **B** and **C** of initial states, the steady-state of the non-Hermitian dynamics takes the form of a Slater determinant describing fermions in a single imaginary-energy band, which is partially filled. This leads to an entanglement entropy scaling as the logarithm of the subsystem size, i.e.

$$S^{B/C}(+\infty) = \frac{1}{3} \ln \ell + \mathcal{O}(1). \tag{35}$$

with a prefactor of the log term equal to $1/3$ as expected for gapless Dirac fermions described by a $(1+1)$D conformal field theory with a central charge $c = 1$ [45]. This result is independent of the state's filling[1]. The scaling of the entanglement entropy is confirmed by the numerical results shown in Fig. 4 for both class **B** and **C**. While the dynamics presents a rather different behavior, the long-time steady-state entanglement converge to the expected scaling with subsystem size, here for a cut $\ell = L/4$.

# 5 Application: the non-Hermitian SSH Hamiltonian

In this section, we examine the non-Hermitian Su-Schrieffer-Heeger (SSH) model [15,17,27, 46–49], which exhibits richer spectral properties than the Hatano-Nelson model, including $\mathcal{PT}$ symmetric and broken phases. We first review the SSH spectral properties and then analyze the steady-state momentum occupation and its connection to the entanglement entropy. In particular, we find that the scaling of the steady-state entanglement entropy depends on the class of the initial state, whether it spontaneously breaks the $\mathcal{PT}$ symmetry, and the filling in classes A and B.

## 5.1 The non-Hermitian SSH Model

The SSH model describes a spinless fermionic chain with 2 atoms per unit cell, giving rise to two sublattices $A, B$, and dimerized hopping $-J \pm h/2$, see Figure 5. We consider a non-Hermitian extension of the SSH model where fermions in the A sublattice are locally amplified - they experience gain - while those in the B sub-lattice are damped - they experience losses. In real-space the non-Hermitian Hamiltonian reads [27,50]

$$\mathcal{H} = -\sum_{l=0}^{L-1} \left[ \left( J - \frac{h}{2} \right) c_{B,l}^\dagger c_{A,l+1} + \left( J + \frac{h}{2} \right) c_{A,l}^\dagger c_{B,l} + \text{h.c.} \right] + i\gamma \sum_{l=0}^{L-1} \left( c_{A,l}^\dagger c_{A,l} - c_{B,l}^\dagger c_{B,l} \right),$$
(36)

which can be casted in the form of our original Eq.(1) and contains now two bands corresponding to the sublattice index $A, B$. In momentum space, it is given by a two band Hamiltonian of the form,

$$\mathcal{H} = \sum_k \Psi_k^\dagger \mathcal{H}_k \Psi_k, \quad \mathcal{H}_k = \begin{pmatrix} i\gamma & J_k \\ J_k^* & -i\gamma \end{pmatrix},$$
(37)

where $J_k = (-J + h/2) e^{-ika} - (J + h/2)$ and $\Psi^\dagger = \begin{pmatrix} c_{k,A}^\dagger & c_{k,B}^\dagger \end{pmatrix}$. The non-Hermitian SSH model is obtained as the no-click limit of a monitored SSH chain, where the local hole density in the A sublattice, $c_{l,A} c_{l,A}^\dagger$, and the local fermionic density in the B sublattice, $c_{l,B}^\dagger c_{l,B}$ are monitored [27].

The Hamiltonian of this non-Hermitian system possesses $\mathcal{PT}$ symmetry, as discussed in subsection 3.2. For each momentum $k$, the single-particle eigenstate of the Hamiltonian can either be an eigenstate of $\mathcal{PT}$ or spontaneously break this symmetry. As previously emphasized, if $\mathcal{PT}$ symmetry is not spontaneously broken, the eigenvalues $\varepsilon_{k,\alpha}$ are real. However, if the symmetry is broken, the spectrum becomes complex. For this specific model, one can show that in this case $\varepsilon_{k,+} = \varepsilon_{k,-}^*$. This can be directly seen from the

---

[1]The difference with the filling is reflected in $\mathcal{O}(1)$ term.

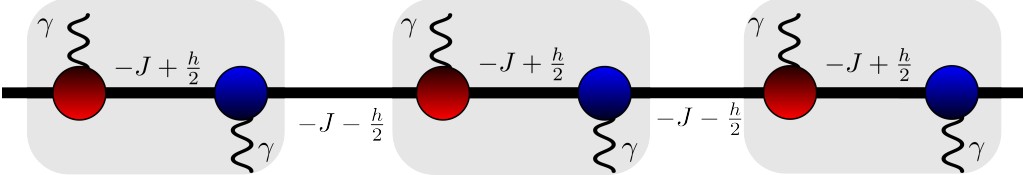

**Figure 5:** Scheme of the non-Hermitian SSH model: Each unit cell is enclosed within a grey area. The intra-cell hopping is given by $-J - h/2$, and the inter-cell hopping is given by $-J + h/2$. The red sites (A sublattice) have a local amplification term, $i\gamma$, while the blue sites (B sublattice) have local damping, $-i\gamma$.

expression for the single-particle dispersion relation

$$\varepsilon_{k,\pm} = \pm\varepsilon_k = \pm\sqrt{h^2 - \gamma^2 + (4J^2 - h^2)\cos^2\left(\frac{k}{2}\right)}. \tag{38}$$

We define three possible phases based on the total number of real eigenvalues:

- The $\mathcal{PT}$-symmetric phase, where the eigenvalues are real for all momenta $k$;

- The $\mathcal{PT}$-fully-broken phase, where the spectrum is purely imaginary for all momenta $k$;

- The $\mathcal{PT}$-mixed phase, where purely real and purely imaginary modes coexist at different momenta $k$.

In most of the literature [24], both the $\mathcal{PT}-$ Fully Broken and $\mathcal{PT}-$ Mixed phases are referred indistinguishably as the $\mathcal{PT}$-broken phase. However, this distinction is important for the remainder of our discussion. The three phases are depicted in the panel $a$) of Fig. 6, where we plot the spectral density of the real eigenvalues defined, in the thermodynamic limit, by the integral $\phi = \int_{-\pi}^{+\pi} dk/2\pi \, \Theta\left(\varepsilon_k^2\right)$, where $\Theta(x)$ is the Heaviside step function.

For $\gamma$ smaller than $h$ and $2J$, the entire spectrum is real and gapped, as seen in panel $b$) of Fig. 6. As $\gamma/J$ increases, the real energy gap decreases and vanishes at $\gamma = h$. At this point, two exceptional points emerge at $k = \pm\pi$, and the system transitions from the $\mathcal{PT}$-symmetric phase to the $\mathcal{PT}$-mixed phase. In this phase, the single-particle energy spectrum is gapless in both its real and imaginary components, as the two bands touch at

$$k_\star = \pm 2\arccos\sqrt{\frac{\gamma^2 - h^2}{4J - h^2}}, \tag{39}$$

for $h < 2J$, as shown in panel $c$) of Fig. 6. For further convenience, we define $k_\star = \pm\pi$ in the $\mathcal{PT}$-symmetric phase and $k_\star = 0$ in the $\mathcal{PT}$-fully-broken phase. With a further increase in $\gamma/J$, another transition occurs at $\gamma_c = 2J$, where the system enters the $\mathcal{PT}$-fully broken phase, characterized by a purely imaginary spectrum with a gap, as seen in panel $d$) of Fig. 6.

## 5.2 Steady-State single-particle distribution

We begin by discussing the steady-state distribution of the single-particle occupation numbers

$$\langle\hat{n}_k\rangle = \sum_k \sum_{\alpha=A,B} \langle\hat{n}_{k,\alpha}\rangle, \tag{40}$$

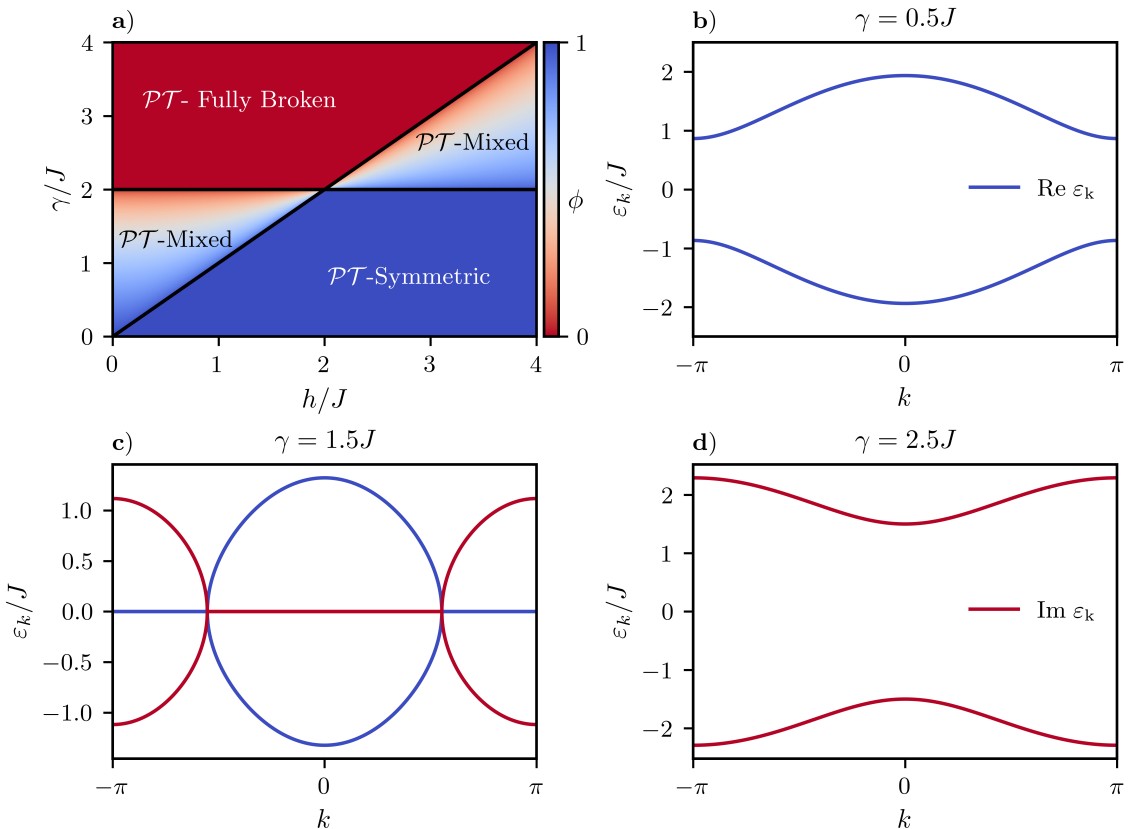

**Figure 6:** (a) Spectral density of the real eigenstates as a function of the system parameters. The transition lines ($\gamma = 2J$ and $h = \gamma$) are marked in black. The single-particle dispersion relation is shown for the $\mathcal{PT}$-symmetric phase in panel (b), the $\mathcal{PT}$-mixed phase in panel (c), and the $\mathcal{PT}$-fully broken phase in panel (d). The blue line corresponds to the real part of the spectrum, while the red line corresponds to the imaginary part. Other parameters: $h = J$.

for different classes of initial states and system parameters. For initial states in class **A**, $\langle \hat{n}_k \rangle$ remains strictly conserved throughout time evolution, regardless of the value of $\gamma/J$. In contrast, for classes **B** and **C**, the distribution is nontrivial. This behavior is illustrated in Fig. 7, where we show the long-time limit of the dynamics for $\langle \hat{n}_k \rangle$ for different values of $\gamma/J$ at fixed $h$, allowing us to examine its dependence across the phase diagram.

In the $\mathcal{PT}$-symmetric phase and for both class **B** and **C** the steady-state occupation resembles strongly the one at initial times, i.e. a uniform distribution in momentum space, with small fluctuations, indicating that all $k$ modes are equally occupied. This can be understood by recalling that in this phase the many-body spectrum is entirely real, meaning that no eigenstate is, in principle, more damped or amplified than the others. This phase can be regarded as a minor perturbation of the Hermitian evolution, preserving the system's dynamical stability. Consequently, $\langle \hat{n}_k \rangle$ is almost conserved, becoming strictly conserved in the limit $\gamma \to 0$.

In the $\mathcal{PT}$-mixed phase and for class **B**, on the other hand, the physics is highly sensitive to the filling of the initial state. The steady state in this regime is constructed by populating the single-particle modes according to the hierarchy of imaginary energies. When the filling is less than the total number of purely imaginary modes, the steady state becomes an eigenstate of $\hat{n}_k$, as illustrated in Fig. 7 for $\nu = 1/8$. In contrast, when the filling exceeds the number of purely imaginary modes, the steady-state distribution

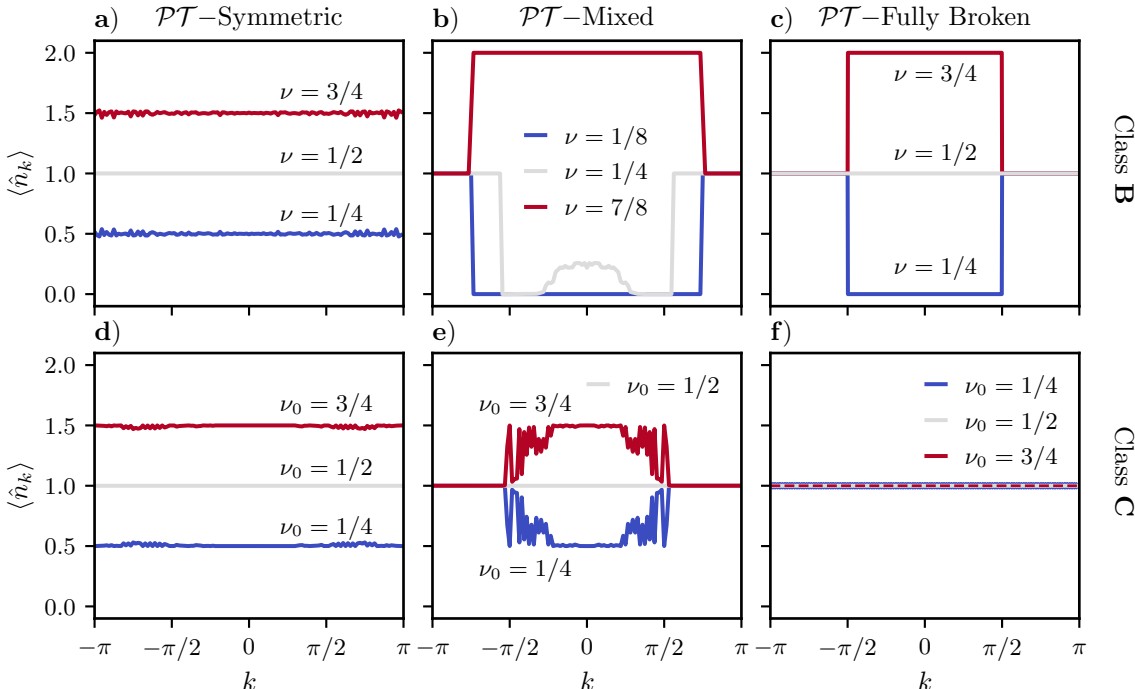

**Figure 7:** Steady-state expectation value of $\hat{n}_k$ in the three spectral regimes for different values of the fillings. The values $\gamma$ are $0.5J$ (panel $a$) and $b$)), $1.5J$ (panel $b$) and $e$)), and $2.5J$ (panel $c$) and $f$)). The upper row corresponds to an initial state in Class **B**, while the bottom row correspond to an initial state in the Class **C**. The color scheme is the same for all three plots in the bottom row. Other parameters: $h = J$.

becomes more complex, as presented for the case of $\nu = 1/4$. Here, the quasi-momentum values $k$ corresponding to purely imaginary energy modes are fully occupied while those associated with real energy modes exhibit a nontrivial $\langle \hat{n}_k \rangle$ distribution, which is harder to predict. We have verified that this distribution, associated with the real modes, depends on the specific choice of the initial state within class **B**.

For class **C** in the $\mathcal{PT}$-mixed phase, the steady-state expectation value of $\hat{n}_k$ is unity for values of $k$ where the spectrum is purely imaginary. In contrast, for $k$ values where the single-particle spectrum remains real, $\langle \hat{n}_k \rangle$ exhibits a nontrivial distribution that depends on the initial state. Figure 7 confirms this behavior, showing that, regardless of the initial expectation value of the filling, the steady state satisfies $\langle \hat{n}_k \rangle = 1$ for all $k$ where the single-particle spectrum is purely imaginary.

Finally, in the $\mathcal{PT}$-fully broken phase, we can apply Eq. (18), leading to a steady state that corresponds to the right eigenstate with the highest imaginary energy, subject to the conservation laws imposed by the initial state. For class **B**, where the total number of particles is conserved, the $k$ modes with the largest imaginary energies are filled up to a value $k_F$ which is determined by the total particle number. This is evident in Fig. 7, where the modes around $k = \pm \pi$, corresponding to the highest imaginary eigenenergies, are occupied first as the total particle number increases.

On the other hand, for class **C**, where the particle number is not fixed, the least damped right eigenstate consists of all $k$ states being occupied by exactly one particle. This results in a steady state where the entire upper imaginary band is populated, as shown in panel $d$) of Fig. 7. Thus, independently of the initial conditions, the steady state in this phase is always half-filled. We note that in both cases of class **B** and **C**, the

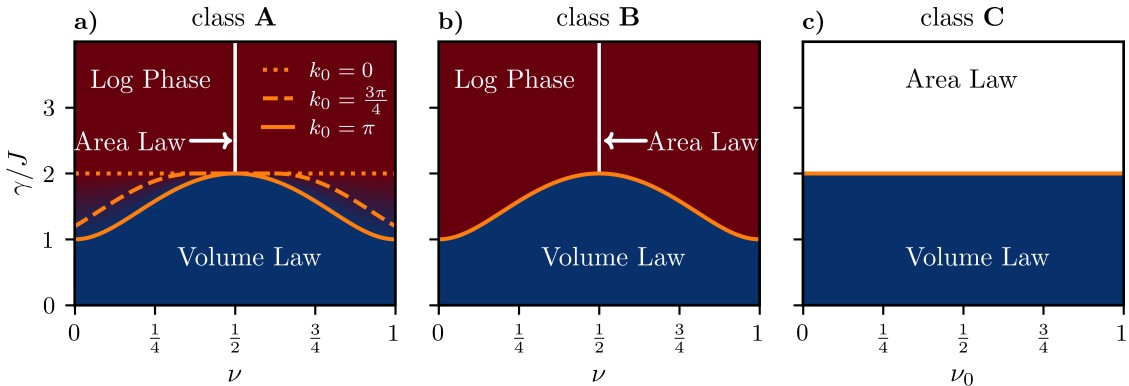

**Figure 8:** Steady-state entanglement phase diagram as a function of $\gamma$ and $\nu$ for the different classes of initial states considered. The blue region indicates a volume-law scaling of the steady-state entanglement entropy, the red region corresponds to logarithmic scaling, and the white region represents area-law scaling. For class **C**, $\nu_0$ denotes the expectation value of the filling in the initial state. The orange lines mark the phase boundaries. In class **A** we observe different boundaries depending on the chosen central $k_0$ from which the state is filled, the boundary is given by Eq. (45). In class **B** the boundary is given by Eq. (46) and is the same as the one for $k_0 = \pm\pi$ in class **A**. The diagram highlights the dominant scaling behavior in the thermodynamic limit. Other parameters: $h = J$.

steady state has restored the symmetry that was broken by the initial state, respectively translation invariance (for class **B**) and conservation of particle number (for class **C**). Our results are therefore consistent and in-line with recent ones exploring symmetry restoration in a related non-Hermitian SSH model [51].

## 5.3    Entanglement Entropy

We now discuss the properties of the steady-state entanglement entropy as a function of system parameters and the class of the initial state. Our main results are summarized in the entanglement phase diagram shown in Fig. 8, as a function of dissipation $\gamma$ and filling $\nu$, for the three different classes of initial states. We briefly highlight its main features, which will be discussed in more detail in the following.

For initial states in class **A** the steady-state entanglement entropy displays either a volume law scaling at weak dissipation or a logarithmic scaling at large dissipation, except for half-filling where the logarithmic phase turns into an area law phase. Due to the conservation of the distribution of $n_k$, the phase diagram in this case strongly depends on the initial state, both in terms of filling $\nu$ and on the offset momentum value $k_0$ used to construct the initial distribution $(n_k)_{t=0}$. As we see in the left panel of Fig. 8, depending on the value of $k_0$ used to construct the initial state, we have different phase boundaries between the volume law and the logarithmic-law phase, including possibly a phase boundary that does not depend on the filling with the exception of the half-filled case where the transition from the volume-law phase is toward an area-law phase.

For initial states in class **B**, the phase diagram also displays three phases characterized, respectively, by a volume law scaling (at weak dissipation), a logarithmic scaling of the entanglement entropy at strong dissipation and a line with area-law scaling for an initial filling equal to $\nu = 1/2$. In contrast to the previous case, the steady-state scaling of the entanglement entropy does not depend on the specific form of the initial states within this class. The boundary of the entanglement transition between the volume law phase and

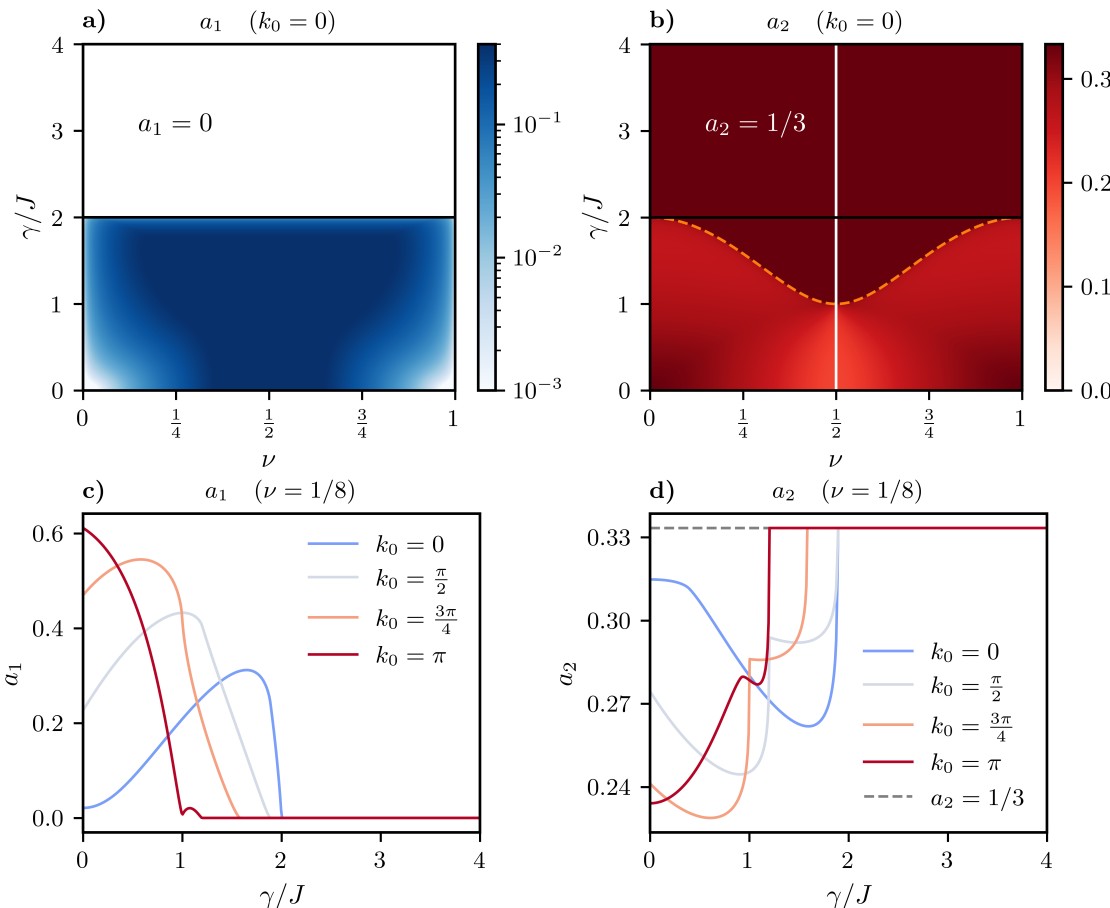

**Figure 9:** *a*) Pre-factor of the volume term in the entanglement entropy for an initial state with $k_0 = 0$. This term vanishes at $\gamma_c = 2J$. *b*) Pre-factor of the logarithmic term for the initial state with $k_0 = 0$. The orange dashed lines indicates the change of $a_2$ into $1/3$. At half-filling the component $a_2$ is always zero. *c*) Pre-factor of the volume term in the entanglement entropy for a fixed filling $\nu = 1/8$ and initial states with different $k_0$. *d*) Pre-factor of the logarithmic term in the entanglement entropy for a fixed filling $\nu = 1/8$ and initial states with different $k_0$. For $\gamma$ large enough we always find $a_2 = 1/3$. Other parameters: $h = J$.

logarithmic law phase exhibits a nontrivial dependence on the filling, making it possible to drive the transition by tuning either the filling or the dissipation.

Finally, in class **C**, the phase diagram features only two phases: a volume-law phase under weak dissipation, corresponding to the $\mathcal{PT}$-symmetric and $\mathcal{PT}$-mixed phases, and an area-law phase under strong dissipation, corresponding to the $\mathcal{PT}$-fully broken phase. These phases emerge independently of the expectation value of the filling in the initial state. We now provide analytical and numerical evidence in support of the general picture drawn so far on the entanglement structure of the non-Hermitian SSH.

### 5.3.1 Class A

For initial states in this class, when the occupation number $\langle \hat{n}_k \rangle$ is conserved and the correlation matrix $G_k^{\alpha\beta}(+\infty)$ diagonal in momentum, it is possible to solve the dynamics and determine the steady-state entanglement entropy analytically, throughout the phase diagram. We note that for this class the specific form of the initial state is crucial, as it

determines and fixes the expectation values of $\hat{n}_k$ for all quasi-momenta $k$. Here we consider initial states defined as Slater determinants of single-particle momentum eigenstates filled from $k_0$. Specifically, we take

$$|\Psi_0\rangle = \prod_{|k-k_0|<2\pi\nu} D_k^\dagger |\text{vac}\rangle, \tag{41}$$

with $|k - k_0|$ defined modulo $2\pi$, and $D_k^\dagger$ is a compact notation to consider the possibility that a given filled momentum $k$ can be single or doubly occupied

$$D_k^\dagger = \begin{cases} \dfrac{c_{kA}^\dagger + c_{kB}^\dagger e^{ik/2}}{\sqrt{2}}, & \text{(single occupancy)} \\[4mm] \left(\dfrac{c_{kA}^\dagger - c_{kB}^\dagger e^{-ik/2}}{\sqrt{2}}\right)\left(\dfrac{c_{kA}^\dagger + c_{kB}^\dagger e^{ik/2}}{\sqrt{2}}\right), & \text{(double occupancy)}. \end{cases} \tag{42}$$

For $k_0 = 0$, this corresponds to the ground-state of the hermitian SSH model ($\gamma = 0$) with $h = 0$. Despite this specific choice of initial state, all subsequent considerations and discussions remain valid and the results can be mapped to other initial distributions.

After the analytical manipulations, that we report in Appendix. C for the interested reader, we obtain two distinct contributions for the steady-state entanglement entropy, besides the irrelevant area law coefficient,

$$S_\ell(+\infty) = a_1 \ell + a_2 \ln \ell + \mathcal{O}(1). \tag{43}$$

As shown in Appendix C, the prefactors $a_1, a_2$, respectively controlling the volume-law and logarithmic-law term, can be computed using the Szegö lemma and the Fisher-Hartwig conjecture [42] through the eigenvalues $\mu_{k,\pm}$ of the steady-state correlation matrix $G_k^{\alpha\beta}(+\infty)$. We obtain the following expression,

$$a_1 = \int_\mathcal{V} \frac{dk}{2\pi} \left[s(\mu_{k,+}) + s(\mu_{k,-})\right], \tag{44}$$

where the integration is performed over the domain $\mathcal{V} = [-k_\star, k_\star] \cap [k_0 - 2\pi\nu, k_0 + 2\pi\nu]$ with $k_\star$ defined in Eq. (39), and the binary entropy function is defined as

$$s(x) = -(1-x)\ln(1-x) - x\ln(x).$$

As obtained from the exact calculation, only the eigenvalues of the correlation matrix, $G^{\alpha\beta}k(+\infty)$, corresponding to a single-occupied quasimomentum $k$ with real energy, contribute to the volume coefficient $a_1$.

Panel $a)$ of Fig. 9 shows the numerical value of $a_1$ for an initial state with $k_0 = 0$. In this case, for all fillings, the coefficient $a_1$ vanishes only when the system transitions to the $\mathcal{PT}$-Fully Broken phase, i.e. for $\gamma = 2J$, which therefore signals the transition out of the volume-law phase for $k_0 = 0$ (see the phase diagram in Fig. 8).

The filling does not affect the transition for this particular initial state ($k_0 = 0$), but in general it does it for other initial states as we show in Fig. 8(a).

In the panel $c)$ of Fig. 9, we show, for a fixed value of the filling $\nu = 1/8$, the dependence of the coefficient $a_1$ with $\gamma$ for initial states with distinct $k_0$. For this filling and $k_0 \neq 0$ we see that $a_1$ vanishes at a dissipation value $\gamma/J < 2$, i.e. the transition out of the volume law phase does not coincide any longer with the full breaking of the $\mathcal{PT}$-symmetry, as was the case for the half-filled SSH model discussed previously [27]. We can derive an expression for the critical value of $\gamma$ at which $a_1$ vanishes, corresponding to the boundary

between the phase with volume-law scaling of the entanglement and the phase exhibiting logarithmic or area-law scaling. The critical value of dissipation reads

$$\gamma_c(k_0, \nu) = \left( h^2 + (4J^2 - h^2) \cos \left[ \frac{\max(|k_0| + \pi|\nu - 1| - \pi, 0)}{2} \right]^2 \right)^{1/2}. \qquad (45)$$

These boundaries are plotted in the panel $a)$ of Fig. 8 for different values of $k_0$, ranging from 0 to $\pi$ (the expression is symmetric in $k_0$). The transition is only independent of the filling when $k_0 = 0$, while for other values, the phase boundary shows a strong dependence on $\nu$, especially away from half-filling.

Returning to the general result in Eq. (43) we see that once $a_1$ vanishes the entanglement entropy scales as the logarithm of the subsystem size, whenever the coefficient $a_2 \neq 0$. Interestingly, this sub-leading term only appears when the system is doped away from half-filling. In fact $a_2$ originates from discontinuities in the two-point correlation matrix $G_k^{\alpha\beta}$ near $k_0 \pm 2\pi\nu$.

The integral yielding $a_2$ is detailed in Appendix C.

In panel $b)$ of Fig. 9, we show the numerical values of $a_2$ for different fillings and $\gamma/J$ for the initial state with $k_0 = 0$. We note that $a_2$ is different from zero throughout the phase diagram, except along the line $\nu = 1/2$, however, its effect is visible only in regions where the leading term $a_1 = 0$. We can understand the origin of the logarithmic phase and its transition into an area law at half-filling from the perspective of our general framework. Indeed, whenever the spectrum is purely imaginary (as for $\gamma/J \geq 2$) the steady state is represented by a Slater determinant obtained by filling the single-particle states with the highest imaginary energies. When the state is away from half filling, we thus get the purely imaginary spectrum to be partially filled, which explains the logarithmic scaling of the entanglement entropy. We also note that the logarithmic contribution to the entanglement entropy features a prefactor of 1/3, as expected from a free-fermionic Conformal Field Theory.

In contrast, when the system is exactly at half-filling, one of the imaginary bands is completely filled while the other remains empty, and the spectrum in that case is effectively gapped. This scenario is analogous to the ground-state of an insulating band model, resulting in an area-law behavior for the entanglement entropy.

Finally, in panel $d)$ of Fig. 9, we show cuts of the coefficient $a_2$ at fixed filling $\nu = 1/8$ and for different initial states parametrized by $k_0$. We can then observe that the volume law to area law transition occurs at a different $\gamma_\star(\nu = 1/4)$ depending on $k_0$. This can be understood from Eq. (44) since $\mathcal{V}$ is based on an intersection that changes depending on $k_0$ and can more easily be null for small and large filling.

### 5.3.2  Class B

For the initial states in class **B**, we investigate the steady-state scaling of the entanglement entropy by propagating the initial state using the Faber polynomial method [37] which is detailed in appendix A. In the following discussion, we fix $h = J$, and tune $\gamma/J$ across the phase diagram in Figure 10 and the filling of the initial state.

In the $\mathcal{PT}$-symmetric phase, that is, for $\gamma/J < \gamma_c(h = 1)/J = 1$, the entanglement entropy scales linearly with the size of the system, $S_\ell \sim \ell$, as shown in panel $a)$ of Fig. 10 for a cut $\ell = L/4$. This result is independent of the system's filling, which controls only the prefactor of the linear term. We can understand this result by noticing that $\mathcal{PT}-$symmetric phase is similar to and adiabatically connected[2] to the unitary evolution, where

---

[2]When tuning $\gamma \to 0$ no gap is open in the single-particle spectrum.

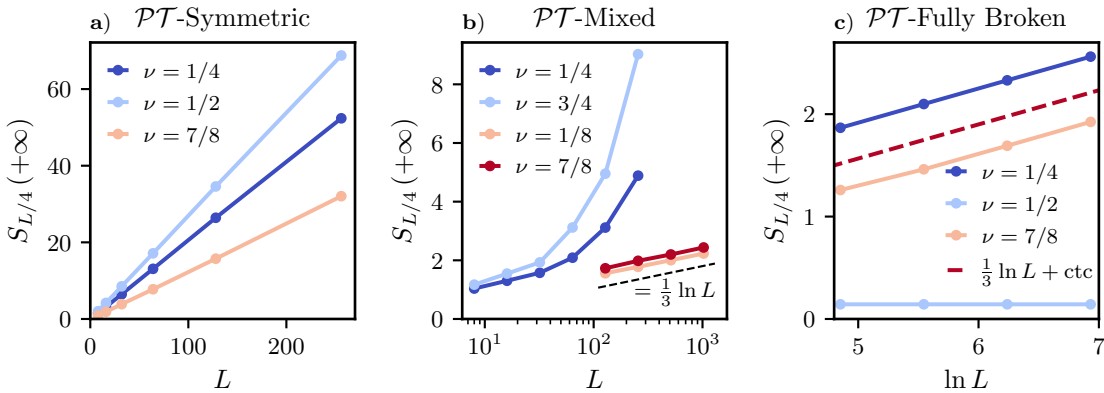

**Figure 10:** Steady-state entanglement entropy for an initial state in the class **B** states in the three different spectral regimes for different of the fillings $\nu$. The values of $\gamma$ are $0.5J$ (a)), $1.5J$ (b)) and $2.5J$ (c)). The parameter $L$ corresponds to the total number of unit cells. Other parameters: $h = J$.

a volume-law scaling of the entanglement entropy is generically observed after a global quantum quench in free fermionic systems [3]. Furthermore, as discussed in Sec. 3, in this phase the system's steady state can be described by a diagonal ensemble, which usually encodes effectively thermal or (generalized) equilibrium states. This further corroborates the volume-law scaling of the entanglement entropy.

Upon increasing the dissipation and entering the $\mathcal{PT}$-mixed phase, volume-law scaling of the entanglement entropy becomes strongly dependent on the initial filling. In particular, we see in panel b) of Fig. 10 that by varying the filling of the initial state, one can tune an entanglement transition from volume scaling to logarithmic scaling. This result can be understood within our general framework: volume-law scaling is observed whenever the state filling exceeds the number of quasimomenta $k$ with purely imaginary energies (see panel b) of Fig. 10) and the modes are quasimomenta $k$ are singly occupied. Conversely, when the filling is less than the number of single-particle modes with purely imaginary energies, the entanglement entropy exhibits logarithmic scaling. A similar behavior occurs if the filling is such that all modes associated with real energies are doubly occupied, as illustrated in the phase diagram of Fig. 8 and seen explicitly for the case $\nu = 7/8$ in Fig. 10.

The results above and in particular the filling-driven entanglement transition between logarithmic and area-law phase are fully consistent, at least at the qualitative level, with our general framework. Technically, however, the existence of a pair of exceptional points prevents us from fully diagonalizing the Hamiltonian throughout the Brillouin zone. Indeed, as discussed in Sec. 5.2 in this $\mathcal{PT}$-mixed phase the steady state is a Slater determinant constructed by occupying the single-particle states with the highest imaginary energies. Consequently, it is possible to apply the Szegö lemma and the Fisher-Hartwig conjecture to corroborate the numerical results shown in Fig. 7.

In the $\mathcal{PT}$-fully broken phase, the spectrum is entirely imaginary, and so we can apply our general framework to make analytical progress. In this scenario, the steady state is described by a Slater determinant formed from the slowest decaying single-particle states. Using the Fisher-Hartwig conjecture, the steady-state entanglement entropy exhibits logarithmic scaling with a $1/3$ prefactor in the gapless phase, while in the specific case of half-filling, it adheres to an area law (see Appendix C for details). This behavior is supported by finite-sized system numerical simulations, as illustrated in Fig. 10. In the thermodynamic limit, the steady state can once again be understood as a Slater determinant, interpreted

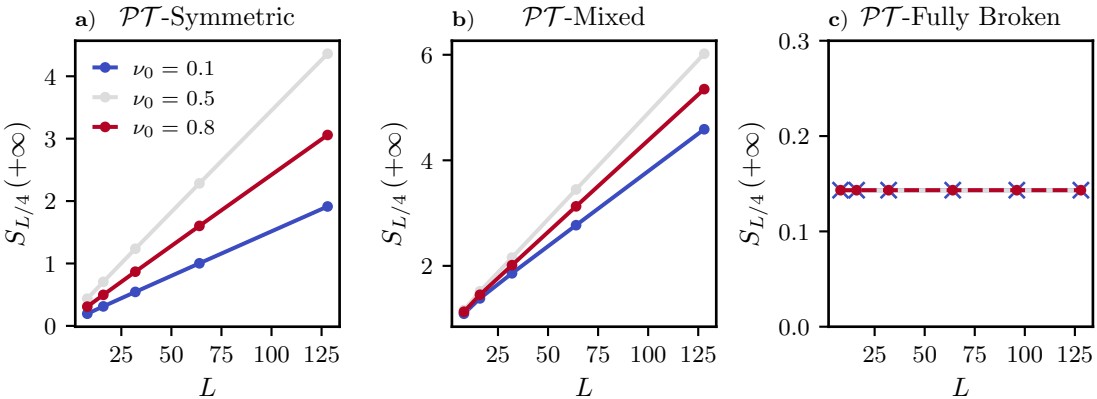

**Figure 11:** Steady-state entanglement entropy for different system parameters for the states in class **C**. The values of $\gamma$ are $0.5J$ ($a$)), $1.5J$ ($b$)) and $2.5J$ ($c$)). Other parameters: $h = J$.

as the low-energy state of a free-fermionic conformal field theory. The prefactor of the logarithmic scaling corresponds to a central charge $c = 1$, whenever $\nu \neq \,^{1}/_{2}$.

We can obtain an analytical expression for the phase boundary in Fig. 8 separating the volume-law phase from the logarithmic-law phase, as a function of the initial filling. This boundary can be determined by examining whether, for a given filling, any quasi-momenta $k$ associated with real energy are singly populated. If this is the case, one obtains a volume law scaling for the entanglement entropy. Conversely, if only purely imaginary modes are occupied and the real modes are empty or doubly occupied, a logarithmic scaling occurs, except for $\nu = 1/2$, which follows an area law scaling. The critical value of $\gamma$ is then given by

$$\gamma_c = \left[ \left( 4J^2 - h^2 \right) \cos^2 \left( \frac{\pi}{2} \left( 1 - 2\nu \right) \right) + h^2 \right]^{1/2} . \tag{46}$$

This is the same boundary found for states considered in class **A** with $k_0 = \pi$. This is the case in class **A** since filling from $k_0 = \pi$ corresponds to filling according to the highest imaginary single-particle energies.

### 5.3.3 Class C

In this last section, we discuss the steady-state entanglement structure for the initial state in class **C**. In Fig. 11, we plot the scaling of the entanglement entropy for a partition $\ell = L/4$ for three values of $\gamma$, respectively in the $\mathcal{PT}$-symmetric, mixed and fully-broken phase and for different values of the expectation value for the initial filling $\nu_0$. In the $\mathcal{PT}$-symmetric and mixed phase, we consistently observe a volume-law scaling of the steady-state entanglement entropy, see Fig. 11(a,b), with the initial filling controlling the slope of the entanglement entropy, i.e. the coefficient of the volume-law term. In contrast, in the fully broken $\mathcal{PT}$ phase, the steady-state entanglement is independent of system size, i.e. the system enters the area law phase. In other words, for initial states in class **C** the phase diagram displays a volume-to-area law entanglement transition, which we locate numerically at $\gamma_c = 2J$, independently on the initial filling $\nu_0$. This value coincides with the spectral transition into the fully broken $\mathcal{PT}$-phase.

The above results are again fully consistent with our general framework for the steady-state occupancy. Indeed, as discussed in Sec. 5.2, for initial states in class **C** the steady state always fills up states with zero imaginary eigenvalue (i.e. purely real eigenvalues)

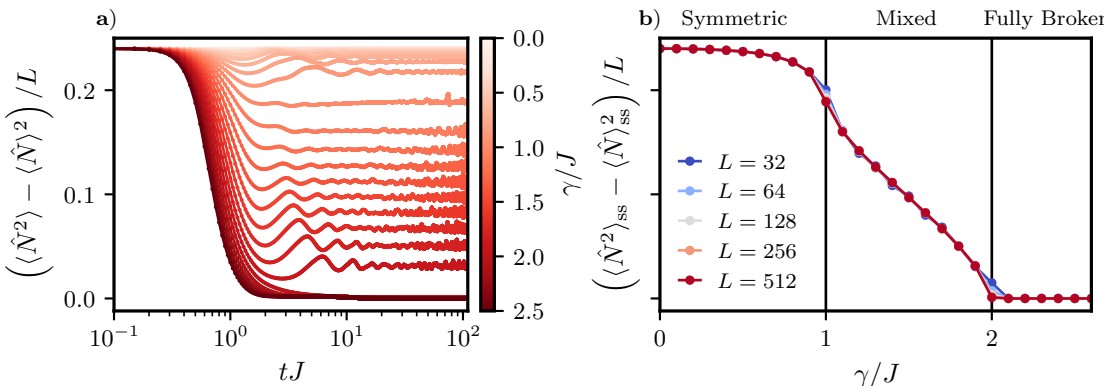

**Figure 12:** Fluctuations of the expectation value of the total particle number for states in class **C** and a system with linear size $L = 512$. *a)* Dynamics of the intensive fluctuations for different values of $\gamma$. *b)* steady state value of the intensive particle number fluctuations as a function of $\gamma/J$ and different linear system sizes. We see that charge-sharpening is not effective for $\gamma < J$ and it is fully complete for $\gamma > 2J$, leaving in between a phase where charge is partially sharpened. In the top of panel *b)*, we identify the corresponding spectral phase for this specific ratios of $\gamma/J$ with $h = J$.

whenever available and therefore the leading contribution to the entanglement entropy displays volume law scaling in the $\mathcal{PT}$ and $\mathcal{PT}$−mixed phases. In the fully broken $\mathcal{PT}$ phase, when the spectrum is complex, the system relaxes into the right eigenstate of the many-body Hamiltonian. As discussed earlier, this corresponds to the Slater determinant obtained by filling all right single-particle states with the highest imaginary energy. Consequently, the correlation matrix is diagonal in the $k$ basis, leading to an area-law scaling of the entanglement entropy. The steady state resembles that of an insulator as it consists of a fully occupied upper imaginary band.

Finally, it is interesting to note that initial states in class **C** which are linear superposition of states with different particle numbers allow to explore charge-sharpening dynamics in a non-Hermitian setting. For monitored random circuits with U(1) symmetry, it is known that a charge-sharpening transition occurs within the volume-law phase [52]. This transition distinguishes a phase where initial charge superpositions are not effectively collapsed by the monitoring protocol, remaining fuzzy along individual trajectories for times $t \sim L$, from a regime where this process occurs rapidly: the system is in a well-defined charge eigenstate well before becoming fully pure. It is therefore tempting to investigate whether a similar behavior occurs within the volume-law phase of our non-Hermitian SSH system, where the charge associated with the U(1) symmetry here is the total particle number. In Fig. 12 *a)*, we plot the dynamics of the fluctuations of the total particle number $\hat{N} = \sum_{i\alpha} n_{i\alpha}$, for different values of $\gamma$. We see that for weak dissipation the fluctuations are almost constant in time and equal to the ones encoded in the initial state. Upon further increasing the dissipation rate, we observe a nontrivial dynamics that reaches a finite steady-state value, i.e. the non-Hermitian evolution has sharpened the charge but not completely. Finally, for large values of $\gamma$ the dynamics collapse the initial superposition and fluctuations at long times vanish. In Fig. 12 *b)*, we plot the steady-state value of the intensive fluctuations, which clearly distinguishes three regimes: for $\gamma < J$ the fluctuations are almost unaffected by the dissipation; for $J < \gamma < 2J$ the residual fluctuations are small and decrease with $\gamma$, and for $\gamma > 2J$ the system at long-times is in eigenstate of the total particle number, that is, the symmetry is restored and the particle-number fluctuations

vanish. Quite interestingly, the three regimes above coincide with the spectral transition where $\mathcal{PT}$-symmetry is first partially broken ($\gamma = J$) and then fully broken ($\gamma = 2J$). This can be understood within our framework: once $\mathcal{PT}$-symmetry breaks, the system develops purely imaginary eigenmodes. These will be populated at long times, reducing the fluctuations of the occupation numbers $n_k$, and, consequently, of the total particle number. This process continues upon increasing $\gamma$ and completes once $\mathcal{PT}$ is fully broken, at which point the charge fluctuations completely vanish.

Although our simple model of non-interacting fermions is not able to reproduce the full features of the charge sharpening criticality observed in random circuits, in particular the different behavior of the time-scales controlling sharpening and purification, it is already quite remarkable that such phenomenology is observed in the no-click limit of purely non-Hermitian evolution. This is even more so considering that our model is non-interacting: for comparison, non-interacting monitored fermions with U(1) symmetry do not sustain a volume-law phase, and so, they also lack a charge sharpening phase. Future work is required to investigate this non-Hermitian charge sharpening transition in a fully interacting and chaotic case.

# 6   Conclusions

In this work, we have introduced a general theoretical framework to understand the steady-state structure of non-interacting fermionic non-Hermitian lattice models and their entanglement content. We have highlighted in particular the role of symmetries and conserved quantities in constraining the dynamics. A unique feature of non-Hermiticity which directly arises from the nonlinearity of measurement back-action is that conserved quantities arise from an interplay between Hamiltonian symmetries and initial states breaking or preserving such symmetry. Focusing on translation-invariant free fermionic models with U(1) symmetry, we have identified three classes of initial states (Class **A**, **B**, and **C**) which break at different levels the symmetries of the problem and therefore have a strong impact on the steady-state properties.

We have shown that for generic non-interacting non-Hermitian Hamiltonian with complex spectrum, the steady state can be obtained by filling single-particle states with the highest imaginary eigenvalue, compatibly with the constraints introduced by the initial states. Initial states in class **A** lead to a dynamic that conserves the average occupation $\langle \hat{n}_k \rangle$, leaving strong imprints on the structure of the steady state. Initial states in class **B** are characterized by a conserved total number of particles, and as a result, the initial filling plays a key role in shaping the steady state. In particular, we have shown that for initial states in this class, the steady state can feature either partially or completely filled bands of imaginary eigenvalues. Finally, the initial states in class **C** are characterized by breaking the particle number conservation, leading to a filling of the imaginary eigenvalue bands without additional constraints. Interestingly, while the initial states break the symmetries of the non-Hermitian Hamiltonian, the dynamics restores them in the steady state. Since for fermionic gaussian states the knowledge of the correlation matrix determines the structure of the entanglement entropy, our results allow to conclude that for non-Hermitian fermions with complex energy spectrum, the entanglement scaling in the steady-state is either area law, if the imaginary eigenvalue bands are gapped and half-filled, or scaling with the logarithm of system size for partially filled bands. This immediately implies that the structure of the steady-state entanglement can be strongly dependent on filling, a feature that is remarkably different from the unitary case and resembles the case of ground-state problems.

We have also discussed the situation in which, due to $\mathcal{PT}$-symmetry, the energy spectrum is either fully real ($\mathcal{PT}$-symmetric phase) or partially real and partially complex (so-called $\mathcal{PT}$ mixed phase). In the first case, we have argued that the steady-state density matrix can be described in terms of a diagonal ensemble, and due to the purely real-spectrum, the entanglement entropy shows volume law scaling. In the $\mathcal{PT}$ mixed phase, the steady-state occupation depends nontrivially on filling: if the purely real energy is partially occupied, the entanglement entropy still shows a volume law scaling.

We have applied our framework to two relevant models of non-Hermitian fermionic lattices, the Hatano-Nelson and the Su-Schrieffer-Heeger models. In the former case, we have shown how the dynamics of single-particle states always leads to partially filled bands, giving rise to a steady-state entanglement that is logarithmic in the system size as in critical systems. For the SSH model, the landscape is richer because of its spectral properties, involving $\mathcal{PT}$ symmetry and its breaking. This has direct consequences on the structure of the steady-state entanglement. In particular, we have shown that in the $\mathcal{PT}$-symmetric phase the dynamics for all initial states resembles the unitary evolution. In particular, local observables thermalize to a diagonal ensemble, and the entanglement entropy displays a volume-law scaling, independently of the initial state.

On the other hand, when the dissipation is large enough such that $\mathcal{PT}$-symmetry is fully broken and the spectrum is purely imaginary, the initial state strongly affects the dynamics. We have shown in Sec. 3 that the steady state is obtained by filling single-particle orbitals with the highest imaginary eigenvalue, compatibly with the initial filling for states in class **B**, or independently of the filling for class **C**. The entanglement entropy is therefore either area law or logarithmic. In contrast, in the $\mathcal{PT}$-mixed phase, the situation is richer for classes **A** and **B**, with a steady-state occupation that depends nontrivially on filling. As such, the entanglement entropy shows a transition driven by filling between logarithmic scaling and volume law scaling. However, for initial states in class **C** in the $\mathcal{PT}$-mixed phase, the entanglement entropy still shows the volume law. Nevertheless, we have unveiled a nontrivial dynamics of the fluctuations of the total particle number which displays a phenomenology which resembles the charge-sharpening of random circuits. To summarize, our results for the non-Hermitian SSH show that by combining conservation laws, $\mathcal{PT}$-symmetry, and the initial state's filling one can generate rich entanglement patterns.

Our work opens up several directions for further investigation. Relaxing the assumption of translation invariance and periodic boundary conditions could allow to discuss entanglement patterns in disordered free fermionic systems or in systems with open boundary conditions, where the skin-effect is expected to play an important role. It would be interesting to explore whether our framework could be generalized to these cases as well. Similarly, bosonic non-Hermitian systems have been shown to display rich physics and could represent another arena to explore the connection between conserved quantities, steady-state occupations, and entanglement. Finally, the major challenge is represented by interacting non-Hermitian many-body systems, which would be exciting to address using extensions of our theoretical framework.

## Acknowledgments

**Funding information**   R.D.S acknowledges funding from Erasmus+ (Erasmus Mundus program of the European Union). We acknowledge Collège de France IPH cluster where the numerical calculations were carried out.

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

# A  Numerical Simulation of Non-Hermitian Lattices

In this appendix, we describe the Faber Polynomial method [37] that we use throughout this work to numerically perform the time-evolution of the states in classes **B** and **C**. In practice, the time-evolution operator in Eq. (5) is expanded in a Faber Polynomial series,

$$
e^{-i\mathcal{H}_{\text{eff}}t} \left| \Psi \left( t_0 \right) \right\rangle = \sum_{n=0}^{+\infty} c_n \left( t \right) F_n \left( \frac{\mathcal{H}_{\text{eff}}}{\lambda} \right) \left| \Psi \left( t_0 \right) \right\rangle ,
$$

$$
c_n \left( t \right) = e^{-i\lambda\gamma_0 t} \left( \frac{-i}{\sqrt{\gamma_1}} \right)^n J_n \left( 2\sqrt{\gamma_1}\lambda t \right) ,
$$

(47)

where $F_n \left( \cdot \right)$ is the n$^{\text{th}}$ Faber Polynomial, $J_n \left( \cdot \right)$ is the n$^{\text{th}}$ Bessel function of the first kind, $\lambda$ is a rescaling parameter and $\gamma_0$ and $\gamma_1$ are parameters related with the bounds of the spectrum of the non-Hermitian Hamiltonian, consult [37] for the specific details. In practice, the series is truncated up to a finite order, and the action of $F_n \left( \mathcal{H}_{\text{eff}}/\lambda \right) \left| \Psi \left( t_0 \right) \right\rangle$ is done efficiently by using the recurrence relation satisfied by the Faber Polynomials, namely,

$$
\begin{aligned}
\left| \Psi_0 \right\rangle &= \left| \Psi \left( t_0 \right) \right\rangle , \\
\left| \Psi_1 \right\rangle &= \left( \tilde{\mathcal{H}}_{\text{eff}} - \gamma_0 \right) \left| \Psi_0 \right\rangle , \\
\left| \Psi_2 \right\rangle &= \left( \tilde{\mathcal{H}}_{\text{eff}} - \gamma_0 \right) \left| \Psi_1 \right\rangle - 2\gamma_1 \left| \Psi_0 \right\rangle , \\
\left| \Psi_{n+1} \right\rangle &= \left( \tilde{\mathcal{H}}_{\text{eff}} - \gamma_0 \right) \left| \Psi_n \right\rangle - \gamma_1 \left| \Psi_{n-1} \right\rangle , \quad n > 2,
\end{aligned}
$$

(48)

where $|\Psi_n\rangle = F_n\left(\mathcal{H}_{\text{eff}}/\lambda\right)|\Psi(t_0)\rangle$. This approach is highly memory efficient, as it requires storing only the last two vectors in memory, $|\Psi_{n-1}\rangle$ and $|\Psi_n\rangle$, to compute the next one, $|\Psi_{n+1}\rangle$. After each time step, the state is normalized. In practice, the expansion in Eq. (47) is done at the level of the single-particle Hamiltonian.

For the states in class **B**, as these have a well-defined particle number M, the many-body state can always be represented in the form

$$|\Psi(t)\rangle = \prod_{n=0}^{\text{M}-1}\left[\sum_{l=0}^{\text{L}-1}\text{U}_{ln}(t)\,c_l^\dagger\right]|\text{vac}\rangle, \tag{49}$$

with M the total number of particles and L the dimension of the single-particle Hilbert space. The time evolution is then reduced to the following equation

$$\mathbf{U}_n = e^{-it\boldsymbol{h}}\,\mathbf{U}_n, \tag{50}$$

where $\mathbf{U}_n$ is the $n^{th}$ column vector and $\boldsymbol{h}$ is the single-particle Hamiltonian matrix, a $\text{L}\times\text{L}$ matrix. In this case, we proceed by expanding $\exp(-it\boldsymbol{h})$ in a Faber series as described above.

The Faber polynomial method can also be used to deal with the states in class **C** [53]. In this case, the fermionic state is parameterized according to

$$|\Psi(t)\rangle = \mathcal{N}(t)\exp\left(-\frac{1}{2}\sum_{m,n}\left[\left(U_t^\dagger\right)^{-1}V_t^\dagger\right]_{m,n}c_m^\dagger c_n^\dagger\right)|\text{vac}\rangle, \tag{51}$$

where $\mathcal{N}$ enforces the correct normalization and $U$ and $V$ are $\text{L}\times\text{L}$ matrices, where L is the dimension of the single-particle Hilbert space. These matrices evolve according to,

$$\begin{pmatrix}U_t\\V_t\end{pmatrix} = e^{-2i\mathbb{H}_{\text{eff}}}\begin{pmatrix}U(0)\\V(0)\end{pmatrix}, \tag{52}$$

where $\mathbb{H}_{\text{eff}}$ is the single-particle non-Hermitian Hamiltonian in the Nambu representation. So, for the states in class **C**, we develop $\exp(-it\mathbb{H}_{\text{eff}})$ in a Faber series as described above.

## B    Diagonal Ensemble

In this Appendix, we derive the diagonal ensemble of Eq. (25). We demonstrate that, analogous to Hermitian systems, one can construct a diagonal ensemble that accurately describes the long-time behavior of the expectation value of a local observable $\mathcal{O}$ in non-Hermitian models. Importantly, this construction is valid only for non-Hermitian systems with a real spectrum. So, it should be applicable to systems driven by $\mathcal{PT}-$ symmetric Hamiltonians in the $\mathcal{PT}-$ symmetric phase, or to generic pseudo-Hermitian Hamiltonians.

To construct the diagonal ensemble, we first perform an eigen-decomposition of the Hamiltonian in a bi-orthogonal basis,

$$\mathcal{H} = \sum_n E_n\left|\Psi_n^R\right\rangle\left\langle\Psi_n^L\right|, \tag{53}$$

where $\left|\Psi_n^{R/L}\right\rangle$ is the many-body right/left $n^{th}$ eigenstate and $E_n \in \mathbb{R}$ the associated eigenenergy. We processed by expressing the long-time expectation value of the observable,

$$\overline{\langle\mathcal{O}\rangle} = \lim_{\tau\to+\infty}\frac{1}{\tau}\int_0^\tau dt\,\langle\Psi(t)|\,\mathcal{O}\,|\Psi(t)\rangle. \tag{54}$$

using the eigendecomposition of the Hamiltonian,

$$\overline{\langle\mathcal{O}\rangle} = \sum_{n,m} \lim_{\tau\to+\infty} \frac{1}{\tau}\int_0^\tau dt\, \frac{\langle\Psi(0)|\,\Psi_n^L\rangle\,\langle\Psi_m^L|\,\Psi(0)\rangle\,\langle\Psi_n^R|\,\mathcal{O}\,|\Psi_m^R\rangle}{\mathcal{N}^2(t)} e^{i(E_n-E_n)t}, \qquad (55)$$

where the normalization factor in the denominator is equal to

$$\mathcal{N}^2(t) = \sum_{n,m}\langle\Psi(0)|\,\Psi_n^L\rangle\,\langle\Psi_m^L|\,\Psi(0)\rangle\,\langle\Psi_n^R\,|\,\Psi_m^R\rangle\, e^{i(E_n-E_m)t}, \qquad (56)$$

where the overlap $\langle\Psi_n^R\,|\,\Psi_m^R\rangle$ is generically non-vanishing for $n\neq m$. In the case of the Hailtonian of the type of Eq. (1), the overlap between many-body eigenstates with different quantum numbers occurs only for a very small subset of states. This is particularly evident in the one-body sector, where each right eigenstate $|\Psi_n^R\rangle$ has a non-zero overlap with precisely $n-1$ other right eigenstates, with $n$ the total number of bands.

Assuming that we can perform the time-average of the norm and the state independently, we obtain the following expressing for the long-time limit of the observable

$$\overline{\langle\mathcal{O}\rangle} = \frac{1}{\mathcal{N}^2}\left[\sum_\alpha \left|\langle\Psi(0)|\,\Psi_\alpha^L\rangle\right|^2\,\langle\Psi_\alpha^R|\,\mathcal{O}\,|\Psi_\alpha^R\rangle + \right.$$
$$\left. + \lim_{\tau\to+\infty}\sum_{\alpha\neq\beta}\langle\Psi(0)|\,\Psi_\alpha^L\rangle\,\langle\Psi_\beta^L|\,\Psi(0)\rangle\,\langle\Psi_\alpha^R|\,\mathcal{O}\,|\Psi_\beta^R\rangle\,\frac{e^{i(E_\alpha-E_\beta)\tau}-1}{i\tau(E_\alpha-E_\beta)}\right]. \qquad (57)$$

Assuming that phase coherence between the different exponential contributions is lost, the second term in the equation should decay to zero in the long-time limit. Consequently, the expectation value reduces to a weighted average of the observable evaluated in the right eigenstates of the Hamiltonian,

$$\overline{\langle\mathcal{O}\rangle} = \frac{1}{\mathcal{N}^2}\sum_\alpha \left|\langle\Psi(0)|\,\Psi_\alpha^L\rangle\right|^2\,\langle\Psi_\alpha^R|\,\mathcal{O}\,|\Psi_\alpha^R\rangle, \qquad (58)$$

where $\mathcal{N}^2 = \sum_\alpha \left|\langle\Psi(0)|\,\Psi_\alpha^L\rangle\right|^2$. So the density matrix predicted from this construction is given by

$$\rho_{DE} = \frac{1}{\mathcal{N}^2}\sum_\alpha \left|\langle\Psi(0)|\,\Psi_\alpha^L\rangle\right|^2\,|\Psi_\alpha^R\rangle\,\langle\Psi_\alpha^R|. \qquad (59)$$

This result is similar to the one derived for an Hermitian model [5]. The main difference lies in the existence of the right and left eigenstates.

## B.1   Testing the Diagonal Ensemble

In this subsection, we make a comparison between the results predicted by the diagonal ensemble $\rho_{DE}$, with those given by the exact numerical simulation of the lattice mode. We take the SSH Hamiltonian of Eq. in the $\mathcal{PT}$- symmetric regime. In this comparison, we only consider observables that are local in real space degrees of freedom such as the onsite particle density or the two-point correlation function between neighboring sites. We expect only local observables to relax, as the system as a whole remains pure at all times. Only the reduced density matrix describing a given small region $A$, $\rho_A = \text{Tr}_{\bar{A}}(|\Psi(t)\rangle\langle\Psi(t)|)$, is expected to reach a steady state that can be described by the diagonal ensemble,

$$\lim_{t\to+\infty}\rho_A(+\infty) = \rho_{DE}. \qquad (60)$$

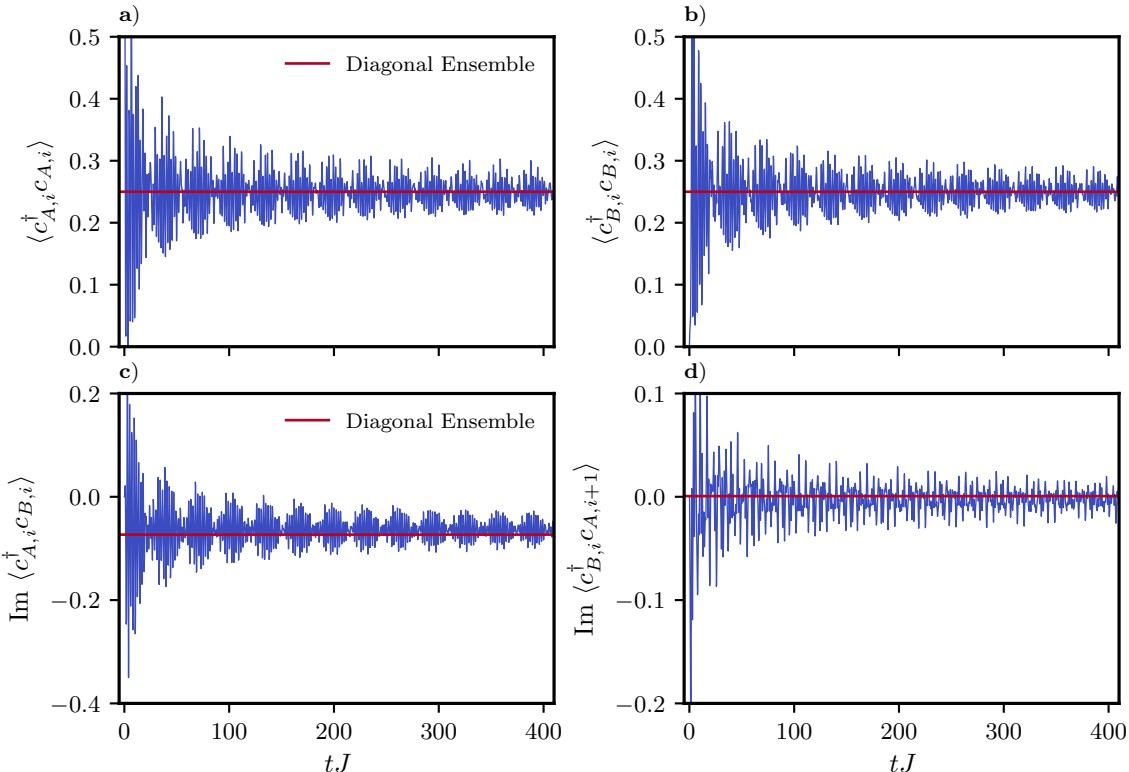

**Figure 13:** Comparison of the expectation value predicted by the diagonal ensemble with the exact numerical evolution for a system of size $L = 512$ unit cells. The effects of finite system size are evident in the small oscillations around the steady-state value. Other parameters: $\gamma = 0.5J$ and $h = J$.

We also assume that the initial states that is of the form,

$$|\Psi_0\rangle = \sum_{l=0}^{N-1} \psi_l \, c_l^\dagger \, |\text{vac}\rangle, \tag{61}$$

with $\sum_l |\psi_l|^2 = 1$. In this way, the initial state has a significant overlap with a large number of left eigenstates of the Hamiltonian.

In Figure 13, we compare the expectation values of the two-point evolving fermionic operators within each unit cell and between adjacent cells considering the initial state $|\Psi_0\rangle = \prod_{n=0}^{L/2} c_{2n,A}^\dagger |0\rangle$. Our results indicate that the diagonal ensemble accurately captures the steady-state value of this correlation.

## C   Analytical Calculation of the Entanglement Entropy

In this appendix, we explain how to calculate the entanglement entropy using the Szegö lemma [3] and the Fisher-Hartwig conjecture [42].

For fermionic Gaussian states, the entanglement entropy can be computed using the two-point correlation matrix truncated to the sites within the region of interest. The entanglement entropy $S_\ell$ of a region $\ell$ is given by the expression

$$S_\ell = \frac{1}{2\pi i} \lim_{\varepsilon \to 0^+} \oint_\Gamma d\lambda \, s\left(\varepsilon, \lambda\right) \frac{d}{d\lambda} \ln \det \mathcal{D}_\lambda^\ell, \tag{62}$$

with

$$\mathcal{D}_\lambda^\ell = \lambda \mathcal{I}_\ell - \mathcal{G}_\ell, \quad \text{and} \quad s(\varepsilon, x) = -(x + \epsilon) \ln(x + \epsilon) - (1 + \epsilon - x) \ln(1 + \epsilon - x),$$

where $\lambda \in \mathbb{C}$, $\mathcal{I}_\ell$ is the identity matrix corresponding to the region $\ell$, and $\mathcal{G}_{i,j}|_{i,j \in \ell} = \langle c_{i\alpha}^\dagger c_{j\beta} \rangle$ is the real-space correlation matrix truncated to $\ell$. The integral is performed along a contour $\Gamma$, which encircles the interval $[0, 1]$ and avoids the branch cuts $(-\infty, -\varepsilon) \cup (1 + \varepsilon, +\infty)$ [42].

In general, calculating the determinant of $D_\lambda^\ell$ analytically is challenging. However, analytical progress can be done if the correlation matrix is a Toeplitz matrix, where $\mathcal{G}_{n,m}^{\alpha\beta} = \mathcal{G}_{n-m}^{\alpha\beta}$, and thus it is diagonal in the momentum sector:

$$\mathcal{G}_{n,m}^{\alpha\beta} = \int_{-\pi}^{+\pi} \frac{dk}{2\pi} e^{ik(n-m)} G_k^{\alpha\beta}. \tag{63}$$

In this case, for $\ell \gg 1$, the determinant $D_\lambda$ can be calculated using the Szegö lemma and the Fisher-Hartwig conjecture:

$$\ln \det \mathcal{D}_\lambda^\ell \underset{\ell \to \infty}{=} \ell \int_{-\pi}^\pi \ln \det \mathcal{D}_k \frac{dk}{2\pi} + \frac{\ln \ell}{4\pi^2} \sum_{r=1}^R \text{Tr} \left[ \ln \left( \mathcal{D}_{k,r}^- \left( \mathcal{D}_{k,r}^+ \right)^{-1} \right) \right]^2 + \mathcal{O}(1), \tag{64}$$

where $\mathcal{D}_{\lambda,k} = \lambda \mathcal{I}_k - G_k$. This identity holds under the assumptions that: (i) $\det \mathcal{D}_k \neq 0$ for all $k$, and (ii) $\det \mathcal{D}_k$ is a piecewise continuous function with jumps at $k = k_1, \ldots, k_r$. The boundary term can also be obtained (see, for instance [54]), but it is not relevant for our purposes.

Combining Eq. (62) with Eq. (64) the entanglement entropy is given as

$$S_\ell = a_1 \ell + a_2 \ln \ell + \mathcal{O}(1), \tag{65}$$

with

$$a_1 = \lim_{\varepsilon \to 0^+} \oint_\Gamma \frac{s(\varepsilon, \lambda)}{4\pi^2 i} \int_{-\pi}^{+\pi} \frac{d}{d\lambda} \ln \det \mathcal{D}_{\lambda,k} \, dk \, d\lambda, \tag{66}$$

$$a_2 = \lim_{\varepsilon \to 0^+} \oint_\Gamma \frac{s(\varepsilon, \lambda)}{8\pi^3 i} \sum_{r=1}^R \frac{d}{d\lambda} \text{Tr} \left[ \ln \left( \mathcal{D}_{k,r}^- \left( \mathcal{D}_{k,r}^+ \right)^{-1} \right) \right]^2 d\lambda. \tag{67}$$

## C.1    The Hatano-Nelson Model

In this subsection, we calculate the steady-state entanglement entropy for an initial state in class **B** and **C** evolving under the Hatano-Nelson model. In these cases, the steady-state correlation matrix is diagonal in the momentum space,

$$\langle c_k^\dagger c_q \rangle_{\text{ss}} = \begin{cases} \delta_{kq}, & k \in [\pi/2 - k_f, \pi/2 + k_f,] \\ 0, & \text{otherwise,} \end{cases} \tag{68}$$

where $k_f$ is defined by the total particle number of the initial state in class **B**, while $k_f = \pi/2$ for the states in class **C**.

The determinant of the matrix $\mathcal{D}_{\lambda,k}$ is discontinuous in $\pi/2 - k_f$ and $\pi/2 + k_f$. Using the Fisher-Hartwig conjecture, we see that there is no volume term, since

$$a_1 = \lim_{\varepsilon \to 0^+} \oint_\Gamma \frac{s(\varepsilon, \lambda)}{4\pi^2 i} \left[ \int_A dk \frac{1}{\lambda - 1} + \int_{\bar{A}} dk \frac{1}{\lambda} \right] = 0, \tag{69}$$

where $A = [\pi/2 - k_f, \pi/2 + k_f,]$ and $\bar{A}$ the complement of $A$ in $[-\pi, \pi]$. In contrast, the coefficient of the logarithmic contribution is non-null and is given by

$$a_2 = -\frac{1}{4\pi^3 i} \lim_{\varepsilon \to 0^+} \oint_\Gamma \frac{d}{d\lambda} s(\varepsilon, \lambda) \ln \left( \frac{\lambda}{\lambda - 1} \right)^2 d\lambda = \frac{1}{\pi^2} \int_0^1 \ln \left( \frac{1 - \lambda}{\lambda} \right)^2 = \frac{1}{3}. \tag{70}$$

## C.2 The SSH Chain

In this subsection, we focus on the non-Hermitian SSH model. We describe the main steps involved in the analytical calculation of the steady-state entanglement entropy for the system parameters for which this is possible in the three different classes.

### C.2.1 Class A

As demonstrated, the expectation value of the operator $\hat{n}_k$ is conserved if and only if the initial state is an eigenstate of the operator. With this in mind, it is possible to explicitly show that the equations of motion satisfied by the two-point correlation matrix factorize for each momentum label. This makes the problem analytically tractable, as the dynamic is confined to the orbital degrees of freedom within each $k$ sector, which in this case are only two. To make it concrete, we assume that the initial state is defined as Eq. (41) of the main text with a filling $\nu$. We start by defining the following basis that simplifies the calculations,

$$
\begin{pmatrix} d_{k,+} \\ d_{k,-} \end{pmatrix} = \frac{1}{\sqrt{2}} \begin{pmatrix} 1 & -e^{ik/2} \\ e^{-ik/2} & 1 \end{pmatrix} \begin{pmatrix} c_{A,k} \\ c_{B,k} \end{pmatrix}, \quad g_k = \begin{pmatrix} \left\langle d^\dagger_{k,+} d_{k,+} \right\rangle & \left\langle d^\dagger_{k,+} d_{k,-} \right\rangle \\ \left\langle d^\dagger_{k,-} d_{k,+} \right\rangle & \left\langle d^\dagger_{k,-} d_{k,-} \right\rangle \end{pmatrix}. \tag{71}
$$

Depending on the eigenvalue $\hat{n}_k = \sum_\alpha d^\dagger_{k,\alpha} d_{k,\alpha}$, which shows if the $k$-mode starts empty, singly occupied, or doubly occupied, we have three initial conditions: $g_k = \mathbb{0}_{2\times2}$, $g_k = (\mathcal{I}_{2\times2} - \sigma_z)/2$, or $g_k = \mathcal{I}_{2\times2}$, respectively. Where $\mathbb{0}_{2\times2}$ is the two by two zero matrix and $\sigma_z$ the $z$ Pauli matrix. After some lengthy algebra using the expression Eq. (7), the dynamics in this basis can be integrated and then rotated back to get the two-point function for a specific $k$ in the original basis [27],

$$
\mathcal{G}_k = \begin{pmatrix} \left\langle c^\dagger_{k,A} c_{k,A} \right\rangle & \left\langle c^\dagger_{k,A} c_{k,B} \right\rangle \\ \left\langle c^\dagger_{k,B} c_{k,A} \right\rangle & \left\langle c^\dagger_{k,B} c_{k,B} \right\rangle \end{pmatrix}. \tag{72}
$$

The average value in the steady state is obtained by using a stationary phase approximation, which gives the following results:

- For $\langle \hat{n}_k \rangle = 0$, $\mathcal{G}_k = \mathbb{0}_{2\times2}$.

- For $\langle \hat{n}_k \rangle = 1$, the correlation matrix is nontrivial and given if $\varepsilon_k^2 \geq 0$ by

$$
\mathcal{G}_k = \frac{1}{2} \begin{pmatrix} 1 & e^{-ik/2} \\ e^{ik/2} & 1 \end{pmatrix} + \begin{pmatrix} 0 & e^{-ik/2}(a_k + 2Jib_k)\chi_k \\ e^{ik/2}(a_k - 2Jib_k)\chi_k & 0 \end{pmatrix}, \tag{73}
$$

where

$$
\chi_k = \frac{A_k}{(1+a_k)A_k} \left( 1 - \frac{1}{\sqrt{2(1+a_k)A_k + 1}} \right), \quad A_k = \frac{\gamma^2 - h^2 \sin(k/2)^2}{2\varepsilon_k^2},
$$

$$
a_k = \frac{\gamma + h\sin(k/2)}{\gamma - h\sin(k/2)}, \quad b_k = \frac{\cos(k/2)}{\gamma - h\sin(k/2)}.
$$

Otherwise if $\varepsilon_k^2 < 0$, we get

$$
\mathcal{G}_k = \frac{1}{2} \begin{pmatrix} 1 & e^{-ik/2} \\ e^{ik/2} & 1 \end{pmatrix} + \frac{1}{2\gamma} \begin{pmatrix} i|\varepsilon_k| & e^{-ik/2}c_k \\ e^{-ik/2}c_k^* & -i|\varepsilon_k| \end{pmatrix}, \tag{74}
$$

with $c_k = \gamma + h\sin(k/2). + 2iJ\cos(k/2)$

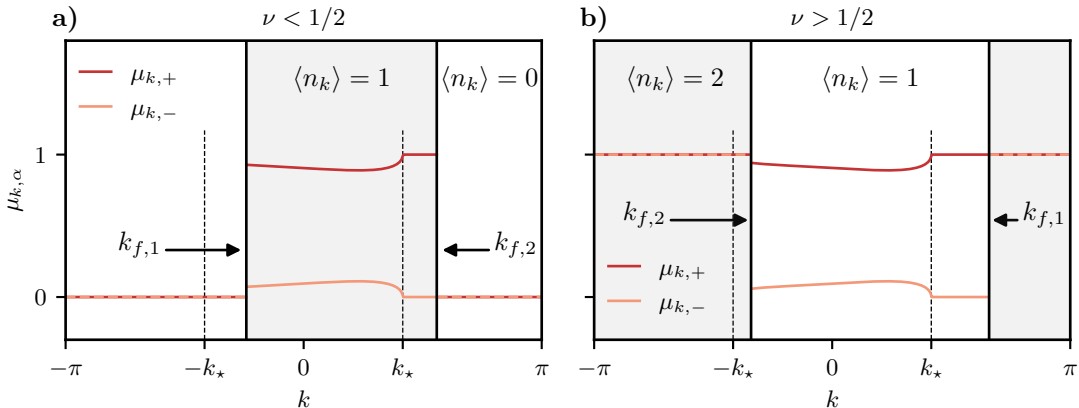

**Figure 14:** Examples of functions $\mu_{k,\pm}$ for $\nu < 1/2$ and $\nu > 1/2$ - a) Case $\nu < 1/2$ where $k$ modes are either empty or singly occupied. Discontinuities are in $k_{f,1} = k_0 - 2\pi\nu$ and $k_{f,2} = k_0 + 2\pi\nu$ and functions are plotted for $\nu = 0.2$ . b) Case $\nu > 1/2$ $k$ modes are either singly or doubly occupied. Discontinuities are in $k_{f,1} = k_0 - 2\pi\nu$ and $k_{f,2} = k_0 + 2\pi\nu$, functions are plotted for $\nu = 0.75$. Other parameters $\gamma = 1.7J$, and $k_0 = 1/2$.

- For $\langle \hat{n}_k \rangle = 2$, $\mathcal{G}_k = \mathcal{I}_{2\times 2}$.

Then we can use the formula Eq. (65) which depends on the eigenvalues $\mu_{k,\pm}$ of $\mathcal{G}_k$ to calculate the coefficient $a_1$ and $a_2$. For $\langle \hat{n}_k \rangle = 0$ or 2, eigenvalues are trivial from $\mathcal{G}_k$, in the case of $\langle \hat{n}_k \rangle = 1$ we obtain

$$
\begin{cases}
\mu_{k,\pm} = \dfrac{1 \pm \xi_k}{2} = \dfrac{1 \pm \sqrt{1 - 4a_k\chi_k + 4(a_k^2 + 4b_k^2)\chi_k^2}}{2}, & \text{if } \varepsilon_k^2 \geq 0, \\[4mm]
\mu_{k,+} = 1, \quad \mu_{k,-} = 0, & \text{if } \varepsilon_k^2 < 0.
\end{cases} \tag{75}
$$

We also define

$$k_{f,1} = k_0 - 2\pi\nu \quad \text{and} \quad k_{f,2} = k_0 + 2\pi\nu, \tag{76}$$

the borders of the fermi sea of the initial state we consider which are depicted in Fig. 14.

Thus $a_1$ is simply computed as shown in the main text, with modes empty or doubly occupied that do not contribute to the linear prefactor. We rewrite the result here for ease,

$$a_1 = \int_{\mathcal{V}} \frac{dk}{2\pi} \left[ s\left(\mu_{k,+}\right) + s\left(\mu_{k,-}\right) \right] , \quad \mathcal{V} = [-k_\star, k_\star] \cap [k_{f,1}, k_{f,2}]. \tag{77}$$

where $s(x) = s(0, x)$ and which directly comes from Eq. (67).

To compute $a_2$, we need to understand the discontinuities of $\mathcal{G}_k$ depending on initial filling $\nu$ and $\gamma$.

First, we describe the situation for $\nu < 1/2$, in this case we have only empty or singly occupied $k$ modes, as shown in Fig. 14 a). We obtain the following two cases :

- If $k_{f,1/2} \in [-\pi, -k_\star] \cup [k_\star, \pi]$ (i.e. in the imaginary part of the spectrum) the discontinuity comes from $\mu_{k,\pm} = 0$ (empty modes) jumping to $\mu_{k,\pm} = 0, 1$ (singly filled modes in the imaginary part of the spectrum). In that case, only one of the eigenvalues is discontinuous and contributes to $a_2$ in Eq. (67).

- If $k_{f,1/2} \in [-k_\star, k_\star]$ (i.e. in the real part of the spectrum) the discontinuity comes from $\mu_{k,\pm} = 0$ (empty modes) jumping to $\mu_{k,\pm} = (1 \pm \xi_k)/2$ (singly filled modes in the real part of the spectrum). Both eigenvalues are discontinuous.

We point out that these are the only discontinuities because for singly filled modes,

$$\mu_{k,\pm}[\varepsilon_k^2 = 0^+] = \mu_{k,\pm}[\varepsilon_k^2 = 0^-]. \tag{78}$$

We now describe the situation for $\nu > 1/2$ as shown in Fig. 14 b). The reasoning is similar; we have two cases:

- If $k_{f,1/2} \in [-\pi, -k_\star] \cup [k_\star, \pi]$ (i.e. in the imaginary part of the spectrum) the discontinuity comes from $\mu_{k,\pm} = 1$ (doubly occupied modes) jumping to $\mu_{k,\pm} = 0, 1$ (singly filled modes in the imaginary part of the spectrum). In that case, only one of the eigenvalues is discontinuous and contributes to $a_2$ in Eq. (67).

- If $k_{f,1/2} \in [-k_\star, k_\star]$ (i.e. in the real part of the spectrum) the discontinuity comes from $\mu_{k,\pm} = 1$ (doubly occupied modes) jumping to $\mu_{k,\pm} = (1 \pm \xi_k)/2$ (singly filled modes in the real part of the spectrum). Both eigenvalues are discontinuous.

We then obtain $a_2$ through Eq. (67) using the discussed discontinuities in matrices $\mathcal{D}_{k,r}^-$ ($\mathcal{D}_{k,r}^+$) for the left (right) limit around the discontinuity[3].

### C.2.2  Class B

For an initial state in class **B** we can compute the entanglement entropy rigorously in the $\mathcal{PT}$ fully broken phase, where Eq. (18) holds. As such, the steady state is a Slater determinant, constructed by filling the single-particle states with the largest imaginary eigenvalues,

$$|\Psi_{ss}\rangle = \frac{1}{\mathcal{N}} \prod_{|k-\pi|<2\pi\nu} \tilde{B}_{k,+}^\dagger |\text{vac}\rangle, \tag{79}$$

where $\mathcal{N}$ is a normalization factor and,

$$\tilde{F}_k^\dagger = \begin{cases} \tilde{f}_{k+}^\dagger, & \text{(single occupancy)}, \\ \tilde{f}_{k-}^\dagger \tilde{f}_{k+}^\dagger, & \text{(double occupancy)}. \end{cases} \tag{80}$$

This fermionic modes appear when diagonalizing the Hamiltonian (Eq.(37)) in a biortogonal basis,

$$\mathcal{H} = \sum_k \left( \varepsilon_{k,+} \tilde{f}_{k+}^\dagger f_{k+} - \varepsilon_{k,-} \tilde{f}_{k-}^\dagger f_{k-} \right), \tag{81}$$

where $f_{k\alpha}$ annihilates a fermion in the left eigenstate, while $\tilde{f}_{k\alpha}^\dagger$ creates a fermion in the right eigenstate of the single-particle Hamiltonian. These operators obey unconventional anticommutation relations:

$$\{f_{q\alpha}, \tilde{f}_{k\beta}^\dagger\} = \delta_{k,q}\delta_{\alpha,\beta}, \quad \{f_{q\alpha}, f_{k\beta}^\dagger\} = \delta_{k,q}V_{\alpha,\beta}, \quad \{\tilde{f}_{q\alpha}, \tilde{f}_{k\beta}^\dagger\} = \delta_{k,q}P_{\alpha,\beta}, \tag{82}$$

with $V_{\alpha,\beta}$ and $P_{\alpha,\beta}$ determined explicitly by the microscopic parameters of the Hamiltonian.

---

[3]We recall that the $\pm$ symbol in $\mu_k$ is due a unit cell of 2 sites, $A$ and $B$, whereas $\pm$ in $\mathcal{D}_{k,r}$ indicates the left and right limits around a discontinuity

Depending on whether the correlation matrix is singly or doubly occupied, it has the eigenvalues:

$$\mu_{k,\pm} = \begin{cases} 0 \text{ and } 0, & (\text{empty}), \\ 0 \text{ and } 1, & (\text{single occupancy}), \\ 1 \text{ and } 1, & (\text{double occupancy}). \end{cases} \tag{83}$$

As explained in the previous subsection, this implies that $a_1 = 0$ and $a_2 = 1/3$.

We note from the numerical results that in the $\mathcal{PT}-$ Mixed phase, the steady-state is also a Slater determinant constructed from the single-particle modes with the highest imaginary energy, whenever the total number of particles is less than the number of purely imaginary eigenmodes of the Hamiltonian. In this case, we can also apply Segzö lemma and the Fisher-Hartwig conjecture, obtaining $a_0 = 0$ and $a_1 = 1/3$.

We stress that we cannot analytically prove that the steady-state is a Slater determinant constructed from the single-particle modes with the highest imaginary energy. In this case, we cannot apply Eq. (18) as the Hamiltonian contains two values of the quasimomentum exceptional points where it cannot be diagoanlized explicitly:

$$\mathcal{H} = \sum_{k \in ]-\pi,\pi[ \setminus \pm k_\star, \alpha} \varepsilon_{k,\alpha} \tilde{f}_{k\alpha}^\dagger f_{k\alpha} + \sum_{k \in \{\pm k_\star\}, \alpha, \beta} h_k^{\alpha\beta} c_{k\alpha}^\dagger c_{k\beta}. \tag{84}$$

### C.2.3   Class C

In the $\mathcal{PT}-$ Fully Broken phase, the steady-state correspondent to an initial state in class **C** is given by filling all the single-particle states with positive imaginary eigenvalues,

$$|\Psi_{ss}\rangle = \prod_{k \in [-\pi, \pi)} \tilde{f}_{k,+}^\dagger |\text{vac}\rangle. \tag{85}$$

Following the same arguments as in the previous subsection, the entanglement entropy follows an area law since the eigenvalues of the correlation matrix for all $k$ are either zero or one, and the determinant of the correlation matrix is continuous. Given this $a_1 = a_2 = 0$.