# Peer review of "Symmetries, Conservation Laws and Entanglement in Non-Hermitian Fermionic Lattices"

_SciPost Physics_

## Round 1 · Referee Report · Anonymous (Referee 1) · 2025-6-18

Report

This paper investigates steady-state entanglement transitions in non-Hermitian quantum many-body systems, with a focus on the role of conservation laws. The authors develop a theoretical framework for translation-invariant, non-interacting fermionic models with U(1) symmetry. They analyze how the spectrum of the non-Hermitian Hamiltonian and conserved quantities shape the steady-state structure and entanglement properties. The framework is applied to two specific models: the Hatano-Nelson and non-Hermitian Su-Schrieffer-Heeger (SSH) models. The results reveal a rich interplay between the spectrum, conservation laws, and entanglement transitions, supported by both analytical and numerical calculations.

The paper addresses an important and timely topic in the study of non-Hermitian quantum systems, advancing our understanding of steady-state entanglement transitions. The results are novel and intriguing, particularly the identification of filling-driven transitions between critical sub-volume scaling and area-law scaling, as well as the volume-law scaling observed for fully real spectra. The paper is well-written, technically sound, and generally of high quality. I believe it makes a significant contribution to the field and recommend it for publication after very minor revisions.

I have one conceptual question that I believe the authors should address: The paper deals with the long-time behavior of non-Hermitian systems, where the non-Hermitian Hamiltonian is justified as arising in the no-click limit of a monitored system. There seems to be a tension here as the no-click limit will become generally very unlikely in the limit of long times. Could the authors comment? On the other hand, how robust do they believe are the results to deviations from the no-click limit, and how might the framework generalize to systems that are described at the level of the quantum master equation so including fluctuations from the environment? (The authors mention Ref 13 by McDonald, Hanai, and Clerk.)

Minor revisions

  1. In several places in Section 5, the authors refer to "amplification" and "damping" of modes. Is this terminology appropriate for fermionic systems? If not, I suggest clarifying or revising the language.

  2. In Appendix B, I noticed a few typos and one missing equation number. These should be corrected for clarity and completeness.

I recommend the paper for publication in SciPost Physics after minor revision. It is well-written, technically sound, and addresses a timely topic, advancing our understanding of non-Hermitian quantum systems significantly.

Recommendation

Publish (surpasses expectations and criteria for this Journal; among top 10%)

  • validity: top
  • significance: high
  • originality: high
  • clarity: top
  • formatting: perfect
  • grammar: perfect

Author:  Rafael Diogo Soares  on 2025-08-08  [id 5711]

(in reply to Report 1 on 2025-06-18)

We thank the Referee for their time spent reviewing our manuscript. The Referee pointed out some questions and requests to be addressed before publication. Here we comment on their questions and requests (following their numbering), pointing out how we accordingly improved our manuscript.

1) The paper deals with the long-time behavior of non-Hermitian systems, where the non-Hermitian Hamiltonian is justified as arising in the no-click limit of a monitored system. There seems to be a tension here as the no-click limit will become generally very unlikely in the limit of long times. Could the authors comment? On the other hand, how robust do they believe are the results to deviations from the no-click limit, and how might the framework generalize to systems that are described at the level of the quantum master equation so including fluctuations from the environment? (The authors mention Ref 13 by McDonald, Hanai, and Clerk.)

  • It remains an open question which features of the no-click limit persist—at least qualitatively—in the general monitored dynamics. In previous works by some of the authors, it was observed that, in certain models, the monitored dynamics closely agree with the no-click limit—for instance, in the Ising model discussed in Ref. 52 of the draft. In contrast, other models, such as the monitored SSH model (also in Ref. 52), exhibit significant deviations. As explored by some of the authors in Ref. 55 of the draft, this behaviour strongly depends on the interplay between the symmetries of the quantum jump operators and those of the underlying model. It would be interesting in future work to investigate whether features of the no-click limit can be preserved by choosing specific jump operators (for example, ones that conserve the nk distribution), and whether the mechanism highlighted in this work remains relevant in such cases, or if the jump operators fundamentally alter the entanglement dynamics. When considering the dynamics of the averaged state, as described at the level of the quantum master equation, there are two points to keep into account to determine how and if our results would generalize. On the one hand, our results rely on the state being pure, whereas the averaged steady state is typically mixed. In such cases, the von Neumann entropy no longer quantifies the entanglement, and it may generically just display volume-law scaling as it is the case for the thermal states and other estimators should be used, such as the negativity. On the other hand it is known that for driven-dissipative quadratic systems the correlation matrix is also fully determined by an effective non-Hermitian Hamiltonian, yet different in general from the no-click limit. This could allow some progress using ideas similar to the ones of our work. We have added a comment on these points in the conclusions.

2) Minor revisions: In several places in Section 5, the authors refer to "amplification" and "damping" of modes. Is this terminology appropriate for fermionic systems? If not, I suggest clarifying or revising the language. In Appendix B, I noticed a few typos and one missing equation number. These should be corrected for clarity and completeness.

  • We have revised the terminology throughout Section 5 and replaced the use of the term “amplification”. Additionally, we have corrected the typos in Appendix B and improved the text for clarity and completeness.

Best regards, Rafael Diogo Soares (on behalf of the authors)

---

## Round 1 · Referee Report · Anonymous (Referee 3) · 2025-6-26

Report

This is an interesting study in the context of measurement induced phase and clarify certain points which are not much discussed in the literature. Therefore I recommend it for the publication.

Recommendation

Publish (meets expectations and criteria for this Journal)

  • validity: ok
  • significance: good
  • originality: ok
  • clarity: good
  • formatting: excellent
  • grammar: excellent

Author:  Rafael Diogo Soares  on 2025-08-08  [id 5710]

(in reply to Report 3 on 2025-06-26)

We thank the Referee for taking the time to review our manuscript and for recommending it for publication.

Best Regards,
Rafael Diogo Soares (on behalf of the authors)

---

## Round 1 · Referee Report · Anonymous (Referee 2) · 2025-6-26

Strengths

  1. Identification of novel phenomena in non-Hermitian systems
  2. Synergy of analytical and computational approaches

Weaknesses

minor points are listed in the report

Report

This is an interesting and timely work that emphasizes the importance of initial states of a non-Hermitian system for their dynamics and entanglement properties. The authors calculated the entanglement entropy in two non-Hermitian models and identified distinct phases that depend on the filling fraction of the initial state. The results of numerical calculations are in agreement with the analytical results obtained within the framework of the Szegö-Fisher-Hartwig approach to asymptotics of determinants of Toeplitz matrices.
I recommend this manuscript for publication with minor points resolved in the resubmission (see requested changes).

Requested changes

  1. The entanglement phases identified in this article are probed numerically for the subsystem size l=L/4. While the analytical formulas are written in terms of the scaling with l, the numerical results do not distinguish between scaling with l and L. For the logarithmic phase, this is of no importance; however, for the volume-law phase, scaling with L for a fixed ratio l/L may produce a spurious contribution. Indeed, imagine that the entanglement entropy contains a subleading term of the form l^2/L that will vanish in the thermodynamic limit L to infinity. However, when scaling with L is considered for l=L/4, this term will produce a volume-law result. Can the scaling with l for the largest L be presented in the manuscript?

  2. Entanglement entropy, while commonly used to describe the entanglement content in the system, contains contribution that are not associated with genuine quantum entanglement. In particular, for systems with the conserved particle number, entanglement entropy also accounts for the ("classical") number entropy. Such non-entanglement contributions to the entanglement entropy can be removed by considering the mutual information; furthermore, genuine entanglement is captured by other entanglement witnesses/measures like concurrence or entanglement negativity (employed by some of the authors in other papers). Importantly, the mutual information can be evaluated for the models considered here just "for free", using the same approach (I guess, other entanglement measures can be obtained in a similar way). It would be interesting to learn whether the volume-law phase as captured by the entanglement entropy indeed corresponds to high entanglement content. I don't insist on evaluating all these entanglement measures here (but strongly encourage the authors to address the mutual information); however, a discussion of these issues should appear in the manuscript to make it even stronger.

  3. Figure 10. What is the reason for the apparent particle-hole asymmetry in panel b (for nu=1/4 and 3/4)? By the way, I would suggest the authors using the same colors for the same values of nu in all panels.

  4. I don't understand the sentence "This result can be understood within our general framework: volume-law scaling is observed whenever the state filling exceeds the number of quasimomenta k with purely imaginary energies (see panel b) of Fig. 10) and the modes are quasimomenta k are singly occupied" on page 24 below Fig. 10 (possibly, a typo).

  5. It seems to me that this reference on a Fisher–Hartwig asymptotic expansion for Toeplitz determinants is sufficiently relevant to the present article for being quoted: https://iopscience.iop.org/article/10.1088/1751-8113/46/8/085003

Recommendation

Publish (easily meets expectations and criteria for this Journal; among top 50%)

  • validity: high
  • significance: top
  • originality: top
  • clarity: high
  • formatting: excellent
  • grammar: excellent

Author:  Rafael Diogo Soares  on 2025-08-08  [id 5712]

(in reply to Report 2 on 2025-06-26)

We thank the Referee for taking the time to review our manuscript. The Referee raised several important questions, which we address below, indicating how we have accordingly improved the manuscript.

1) The entanglement phases identified in this article are probed numerically for the subsystem size $\ell=L/4$. While the analytical formulas are written in terms of the scaling with l, the numerical results do not distinguish between scaling with $\ell$ and$L$. For the logarithmic phase, this is of no importance; however, for the volume-law phase, scaling with $L$ for a fixed ratio $\ell/L$ may produce a spurious contribution. Indeed, imagine that the entanglement entropy contains a subleading term of the form $\ell^2/L$ that will vanish in the thermodynamic limit L to infinity. However, when scaling with $L$ is considered for $\ell=L/4$, this term will produce a volume-law result. Can the scaling with l for the largest $L$ be presented in the manuscript?

  • We have included these results in a new appendix (appendix D), where it is possible to observe that the linear scaling with $L$ is independent of the subsystem size.

2) Entanglement entropy, while commonly used to describe the entanglement content in the system, contains contribution that are not associated with genuine quantum entanglement. In particular, for systems with the conserved particle number, entanglement entropy also accounts for the ("classical") number entropy. Such non-entanglement contributions to the entanglement entropy can be removed by considering the mutual information; furthermore, genuine entanglement is captured by other entanglement witnesses/measures like concurrence or entanglement negativity (employed by some of the authors in other papers). Importantly, the mutual information can be evaluated for the models considered here just "for free", using the same approach (I guess, other entanglement measures can be obtained in a similar way). It would be interesting to learn whether the volume-law phase as captured by the entanglement entropy indeed corresponds to high entanglement content. I don't insist on evaluating all these entanglement measures here (but strongly encourage the authors to address the mutual information); however, a discussion of these issues should appear in the manuscript to make it even stronger.

  • In the case of class B, the total particle number is conserved, and therefore the reduced density matrix can be decomposed into blocks with well-defined particle number. However, this decomposition is not valid for class C, where the state is not an eigenstate of the total particle number. As such, for class B, the entanglement entropy can be decomposed into two contributions: the entropy arising from number fluctuations (number entropy), and the entropy associated with configurations at fixed particle number. We have attached a plot in the response showing that, upon subtracting the number entropy from the total entanglement entropy, the remaining contribution still exhibits a volume law across different fillings. However, we emphasize that in this work, we quantify entanglement with respect to a bipartition of a pure state. In this case, the von Neumann entropy of the reduced density matrix is the only appropriate quantity to measure entanglement across the bipartition, as it is the only measure that satisfies all the necessary axioms required of a proper entanglement measure (see, for example, Rev. Mod. Phys. 81, 865 (2009)). On the other hand, when dealing with mixed states (which is not the case in our work), the von Neumann entropy of the reduced density matrix is no longer a valid entanglement measure, as it cannot distinguish between quantum and classical correlations. In such cases, alternative quantities—such as entanglement negativity, entanglement of formation, etc.—must be considered. However, we stress that these are not proper entanglement measures in the strict sense. For instance, entanglement negativity provides only a sufficient condition for entanglement and may fail to detect it in certain cases, since vanishing negativity does not imply the absence of quantum entanglement (see Phys. Rev. A 65, 032314). As for mutual information: when applied to a bipartition of a pure state, it reduces to twice the von Neumann entropy. However, if one considers the mutual information between two disjoint regions—say, regions A and C in a tripartition of the system into A, B, and C—then the problem reduces to quantifying entanglement in a mixed state, since the joint density matrix $\rho_{A+C}$ is mixed. In this context, mutual information includes both genuine entanglement contributions and additional terms arising from classical correlations due to the fact that the density matrix contains a classical mixture of states, making it unsuitable as a faithful entanglement measure. In fact, it can be shown that mixtures of separable states (which contain no entanglement) can still exhibit non-zero mutual information. We have attached a plot showing the behaviour of the mutual information in this tripartite setting. While it also exhibits volume-law scaling, as we have emphasised, this does not imply volume-law entanglement in the system.

3) Figure 10. What is the reason for the apparent particle-hole asymmetry in panel b (for $\nu=1/4$ and$ 3/4$)? By the way, I would suggest the authors using the same colors for the same values of $\nu$ in all panels.

  • We thank the referee for noticing this, we were plotting the wrong set of data for the $\nu = 3/4$. We have corrected and added the data for $\nu=3/4$ case showing that indeed there is the particle-hole symmetry. We have also improved the color of the plots accordingly.

4) I don't understand the sentence "This result can be understood within our general framework: volume-law scaling is observed whenever the state filling exceeds the number of quasimomenta $k$ with purely imaginary energies (see panel b) of Fig. 10) and the modes are quasimomenta $k$ are singly occupied" on page 24 below Fig. 10 (possibly, a typo).

  • The sentence had a typo. We have corrected the sentence in the main text to: “This result can be understood within our general framework: volume-law scaling is observed whenever the state filling exceeds the number of quasimomenta $k$ with purely imaginary energies (see panel b) of Fig. 10) and the modes with a real energy are singly occupied”

5) It seems to me that this reference on a Fisher–Hartwig asymptotic expansion for Toeplitz determinants is sufficiently relevant to the present article for being quoted: https://iopscience.iop.org/article/10.1088/1751-8113/46/8/085003

  • We have included this reference and improved the references on the Fisher-Hartwig expansion.

Best regards, Rafael Diogo Soares (on behalf of the authors)

Attachment:

mi_delta_s.pdf

---

## Editorial Decision

resubmitted